# Observation of bioaerosol transport using wideband integrated bioaerosol sensor and coherent Doppler lidar

Dawei Tang[1], Tianwen Wei[1], Jinlong Yuan[1], Haiyun Xia[1,2,3,*], Xiankang Dou[1,4]

[1]CAS Key Laboratory of Geospace Environment, School of Earth and Space Science, USTC, Hefei 230026, China
[2]Hefei National Laboratory for Physical Sciences at the Microscale, Hefei 230026, China
[3]CAS Center for Excellence in Comparative Planetology, Hefei 230026, China
[4]School of Electronic Information, Wuhan University, Wuhan 430072, China

*Correspondence to:* Haiyun Xia (hsia@ustc.edu.cn)

**Abstract.** Bioaerosols are usually defined as aerosols arising from biological systems such as bacteria, fungi, and viruses. They play an important role in atmospheric physical and chemical processes including ice nucleation and cloud condensation. As such, their dispersion affects not only public health but regional climate as well. Lidar is an effective technique for aerosol detection and pollution monitoring. It is also used to profile the vertical distribution of wind vectors. In this paper, a coherent Doppler wind lidar (CDWL) is deployed for aerosol and wind detection in Hefei, China, from 11 to 20 March in 2020. A wideband integrated bioaerosol sensor (WIBS) is used to monitor variations in local fluorescent bioaerosols. Three aerosol transport events are captured. The WIBS data show that during these transport events, several types of fluorescent aerosol particles exhibit abnormal increases in either their concentration, number fractions to total particles, and number fractions to whole fluorescent aerosols. These increases are attributed to external fluorescent bioaerosols instead of local bioaerosols. Based on the Hybrid Single Particle Lagrangian Integrated Trajectory (HYSPLIT) backward trajectory model and the characteristics of external aerosols in WIBS, their possible sources, transport paths, and components are discussed. The results prove the influence of external aerosol transport on local high particulate matter (PM) pollution and fluorescent aerosol particle composition. The combination of WIBS and CDWL expands the aerosol monitoring parameters and provides a potential method for real-time monitoring of fluorescent biological aerosol transport events. In addition, it also helps to understand the relationships between atmospheric phenomena at high altitudes like virga and the variation of surface bioaerosol. It contributes to the further understanding of long-range bioaerosol transport, the roles of bioaerosols in atmospheric processes and in aerosol-cloud-precipitation interactions.

## 1. Introduction

Aerosols are suspensions of solid particles or liquid droplets in the atmosphere. Biological aerosols or bioaerosols are atmospheric aerosols derived from biological sources in a broad sense. In a narrow sense, bioaerosols refer to primary biological aerosol particles (PBAP), which means biological material directly emitted to the atmosphere rather than being formed through gas-to-particle conversion. The latter are called Biogenic Secondary Organic Aerosols and are beyond the scope of this paper. Bioaerosols show great diversity in species, including bacteria, fungi, viruses, pollen, algae, and their fragments. Their sizes consequently range from a few nanometers to hundreds of microns. Due to their

hygroscopicity (Petters and Kreidenweis, 2007), bioaerosols show higher efficiency in acting as ice nuclei (IN) and cloud condensation nuclei (CCN) compared with non-organic aerosols. Thus, they have

an important impact on regional climate by participating in atmospheric physical and chemical processes, including cloud formation and precipitation. Bioaerosols are also associated with health problems including infectious, toxic, and hypersensitivity diseases. Some kinds of bioaerosols have the potential to be used for bioterrorism. Therefore, in recent years bioaerosols have been drawing increasing amounts of attention.

45         Due to the diffusion caused by atmospheric turbulence below the planetary boundary layer (PBL), surface aerosol particles are mainly dispersed in the PBL. This dispersion continues until they are removed from the atmosphere by the dry or wet deposition process. However, under specific conditions, a fraction of aerosol particles may be entrained above the PBL and enter the free troposphere. Once there, aerosol particles are no longer affected by the height of the PBL or its daily behavior and can perform

long-range transport under the action of wind or sandstorms until re-entry to the PBL via transport. To survive in dry and intense solar radiation environments at high altitudes during long-range transport, bioaerosols have developed some survival mechanisms including pigment deposition  (Tong and Lighthart, 1997), sporulation (Griffin, 2007), and attaching themselves to other particles like dust (Griffin et al., 2001). These survival mechanisms depend on the type of bioaerosol, for instance, due to their

relatively small size compared with bacteria and fungi, viruses are more likely to be attached to other large pre-existing particles under the influence of Brownian motion.

There are increasing evidences showing the long-range transport capability and pathways of bioaerosols. In Asia, bioaerosols are frequently observed being transported along with dust during Asian dust events. These affect the concentrations and structures of airborne biological communities in the

surrounding area. In Japan, Asian dust events affect airborne bacteria communities in the air near the surface and free troposphere (Hua et al., 2007; Maki et al., 2014, 2015). An increased level of fungal spores is found in Hualien, Taiwan during sandstorms originating from Asian deserts (Ho et al., 2005; Wu et al., 2004). In Korea, Asian dust events impact airborne fungal concentrations and communities (Jeon et al., 2011, 2013). DAPI staining and DNA sequencing results show an increased level of airborne

bacteria concentrations and diversity in Northern China during Asian dust events (Tang et al., 2018). However, these studies are based on aerosol sampling technologies and offline analyses such as microscopy analysis and DNA sequencing. These offline analysis methods limit the temporal resolution during monitoring and the detection time for the external aerosol transport event. These sampling and offline analysis methods provide a typical time resolution ranging from a few hours to a few days, which

may not provide the ability to distinguish the bioaerosol process over a short time scale from the slower bioaerosol long-term variation trend.

To overcome these challenges, several online measurement instruments using light-induced fluorescence (LIF) technologies have been developed(Fennelly et al., 2017) based on the autofluorescent properties of bioaerosol. These instruments monitor fluorescent aerosol particles (FAPs) as a proxy for

bioaerosol particles. The wideband integrated bioaerosol sensor (WIBS, Droplet Measurement Technologies, Inc.) detects the biological fluorescent component of aerosol particles, such as Tryptophan, Riboflavin, or NADH during measurement. It can provide time-dependent characteristics for different

types of FAPs and non-FAPs. However, WIBS still has some limitations in detecting bioaerosols. For example, WIBS can not detect particles whose size is smaller than 0.5 μm, so WIBS has limited potential in viruses detection and focuses on relatively coarse bioaerosols. Besides, non-biological fluorescent components on aerosol particles, such as polycyclic aromatic hydrocarbons (PAHs), humic acids, and fulvic acids may act as interferent during WIBS measurements. In recent years, WIBS is used under different conditions such as at high altitudes (Crawford et al., 2016; Yue et al., 2019), forests, rainforests (Gabey et al., 2011), seashore (Daly et al., 2019), rural areas (Healy et al., 2014), urban areas (Cheng et al., 2020; Ma et al., 2019; Yu et al., 2016; Yue et al., 2016) and in airborne observations (Perring et al., 2015; Ziemba et al., 2016). These measurement campaigns prove its effectiveness and consistency with other offline detection results (Feeney et al., 2018; Fernández-Rodríguez et al., 2018).

Direct real-time observations of aerosol transport help us to understand this process comprehensively. As an active remote sensing tool, lidar can provide long-term and continuous monitoring of aerosol and wind vector vertical profiles with high spatial and temporal resolutions. In recent years, ground-based and satellite-borne lidar combined with other observation data on the ground, such as meteorology, $O_3$, and particulate matter (PM) concentration data, are widely used to study haze or pollution episodes caused by aerosol transboundary transport (Fang et al., 2021; Huang et al., 2021; Qin et al., 2016; Wang et al., 2019b; Yang et al., 2021a, 2021b). However, the above studies are limited to using local PM concentration data to characterize the monitored aerosols, while the variations of different types of aerosols are not characterized.

In this paper, a coherent Doppler wind lidar (CDWL) system is used to detect the vertical profiles of atmospheric aerosols and wind over Hefei, China. A WIBS is used to monitor the concentration and time variation of various types of local fluorescent bioaerosol particles from 11 to 20 March 2020. During the experiment, three aerosol transport events and abnormal changes in some types of fluorescent aerosols are captured. After an analysis combining local meteorological data and Hybrid Single Particle Lagrangian Integrated Trajectory (HYSPLIT) backward trajectory model, these abnormal changes in WIBS data are attributed to transport from external fluorescent bioaerosols. The possible types and origins of external aerosols are discussed.

This paper provides a new perspective for the study of bioaerosol transport. CDWL enables continuous monitoring of multiple atmospheric parameters in real time, such as aerosol extinction coefficient, wind vector, turbulence activity, precipitation, etc. Based on LIF technologies, WIBS provides detailed single-particle information containing up to 5 parameters. It provides a higher time resolution monitoring of aerosols compared with traditional offline measurement methods based on sampling analysis, and compared with online aerosol measurement instruments such as particle sizer, it expands the dimension of aerosol measurement and thus enables categorized monitoring of aerosol. The combination of these two instruments helps to understand the potential impact of external bioaerosols on local bioaerosol composition during aerosol transport. In addition, lidar is capable of detecting atmospheric phenomena in high altitudes which cannot be measured by ground-based in-situ measurement instruments such as the virga and thus enables discovering their relationships with the variation of bioaerosols at ground level during this event. The phenomenon suggests that the combination

of these two instruments also contributes to a further understanding of the role of bioaerosols in atmospheric processes and in aerosol-cloud-precipitation interactions.

In Sect. 2 the instrument parameters, experimental sites, data used, and processing method are introduced. The observation results during each event are discussed in Sect. 3. The conclusions are given in Sect. 4.

## 2. Data and Method

### 2.1. Ground-based lidar measurements

In this study, a CDWL system is utilized. The CDWL emits pulsed laser at a repetition rate of 10 kHz with 100 μJ pulse energy and operates at an eye-safe wavelength of 1.5 μm. Detailed information about the system and its applications can be found in previous work (Jia et al., 2019; Wei et al., 2019, 2020; Yuan et al., 2020).

The lidar observation is performed on the campus of the University of Science and Technology of China (USTC, 31.84 °N,117.26 °E) located in the urban area of Hefei, Anhui Province. The lidar system is placed on the grassland about 100 m west away from the building of the School of Earth and Space Science (SESS). It operates in the velocity-azimuth display (VAD) scanning mode with a fixed elevation angle of 60°. The carrier-to-noise ratio (CNR), Doppler spectral width, and radial wind speed are saved as Level-1 data. Wind vector profiles are then retrieved using the filtered sine wave fitting (FSWF) method (Banakh et al., 2010). In this paper, the wind direction of 0° corresponds to horizontal wind from the north, and the angle increases clockwise. A positive vertical speed value in lidar data indicates a downward direction and otherwise the opposite. The Doppler spectral width is an indicator of turbulence. In addition, it can also be broadened during rainfall events (Wei et al., 2021).

The attenuated backscatter coefficient ($\beta'$) is derived by a semi-qualitative method (Pentikäinen et al., 2020) from CNR and calculated by

$$\beta'(R) = C_0 \, CNR(R) * R^2 / T_f(R) \, , \tag{1}$$

where $C_0$ is a calibration factor containing all technical system parameters (O'Connor et al., 2004) and derived by the integration of the backscattered signal over the optically thick, non-drizzling stratocumulus that can totally attenuate the laser energy. The focus function $T_f(R)$ is retrieved from horizontal scanning results by assuming that the aerosol distribution is homogeneous (Yang et al., 2020). And finally, the $R^2$ is the term of range correction. The effectiveness of this method is demonstrated in Hong Kong, Iceland, and Hefei observation campaigns (Huang et al., 2021; Wei et al., 2022; Yang et al., 2020). The lower threshold of CNR is set to -35 dB for low uncertainty in $\beta'$ retrieval.

### 2.2. PM data and meteorological data

PM$_{2.5}$ and PM$_{10}$ represent particulate matter with aerodynamic diameters less than or equal to 2.5 μm and 10 μm, respectively. At present, many cities in China have established stations with PM$_{2.5}$ and PM$_{10}$ monitoring capabilities, which constitute a part of the National Real-Time Air Quality Reporting System by the China National Environmental Monitoring Center. During the observation in Hefei, hourly PM concentration data published by the Department of Ecology and Environment of Anhui Province

(https://sthjt.ah.gov.cn/) are used to be compared with the lidar observation and surface aerosol concentration measured by WIBS. These data are comprehensive values from the measurement results of multiple stations in Hefei, whose locations are shown in https://aqicn.org/city/hefei/. The nearest station to our lidar is located on Changjiang Middle Road (31.852 °N, 117.25 °E), about 2.7 km northwest of the USTC campus.

The real-time temperature and relative humidity are observed by a weather station (Davis, Wireless Vantage Pro2 Plus). Rainrate is monitored by a laser disdrometer (OTT, Parsivel2) and visibility is measured using a visibility sensor (Vaisala, PWD50). These instruments are located on the rooftop of the SESS building.

### 2.3. WIBS data measurements and processing

The WIBS uses light scattering and fluorescence spectroscopy to detect up to five parameters of every interrogated particle including particle size, asphericity factor (AF), and fluorescence intensity in 3 fluorescent channels. Particle size and AF are derived from the elastic light scattering of aerosol particles irradiated by a 635 nm diode laser, where particle size ranges from 0.5 μm to 30 μm and AF ranges from 0–100. In theory, perfectly spherical particles exhibit an AF value of 0, whereas an AF value close to 100 indicates a fiber-like particle. An elastic light scattering signal beyond the threshold will trigger two xenon flashlamps to emit light at wavelengths of 280 nm and 370 nm, respectively, to excite the fluorescence emission of interrogated particles. The fluorescence signal will be recorded in two wavelength bands (310–400 nm and 420–650 nm). These configurations result in three fluorescence detection channels with different excitation and detection wavelength bands: The FL1 channel with excitation at 280 nm and detection in the 310–400 nm band, the FL2 channel with excitation at 280 nm, and detection in the 420–650 nm band and the FL3 channel with excitation at 370 nm and detection in the 420–650 nm band.

A set of fluorescent thresholds is used to discriminate between fluorescent (marked as 'fluor' in the following section) and non-fluorescent (marked as 'nonfluor' in the following section) aerosol particles. Any particle whose fluorescence intensity on any one of the fluorescence channels (FL1, FL2, or FL3) exceeds its threshold will be regarded as fluorescent. In this paper, the fluorescent threshold $E_{Threshold}$ in each channel is defined as

$$E_{Threshold} = E + 3\sigma \ , \tag{2}$$

Where $E$ is the signal baseline and $\sigma$ is the standard deviation of the signal in each channel. The above two parameters are calculated from the result of the WIBS 'Forced Trigger' (FT) working process. During the FT process, the xenon flashlamps are triggered to fire continuously and fluorescent signals in three channels are recorded in the absence of sampled particles. As mentioned before, non-biological fluorescent components on aerosol particles can be fluorescent interferent during WIBS observation. So, the concentrations of fluorescent aerosol particles can be higher than the actual concentration of local fluorescent biological aerosol particles. Proper fluorescent threshold configuration can eliminate these non-biological interferents as much as possible and remain biological particles categorized into fluorescent as many as possible. Although laboratory test (Savage et al., 2017) shows that applying a higher threshold like 6σ or 9σ threshold can effectively exclude interferent like wood smoke and brown

carbon from being categorized into fluorescent, filed campaign (Yue et al., 2019) suggest that a proportion of bioaerosols can be misclassified into non-fluorescent particles when elevating threshold. The 3σ threshold strategy adopted in this paper is to keep consistent with previous works (Crawford et al., 2015; Yu et al., 2016; Yue et al., 2016).

Using the threshold strategy in Perring et al.(2015), whole fluorescent particles can be further categorized into seven types of fluorescent particles (A, B, C, AB, AC, BC, or ABC). The characteristics of these types of fluorescent particles and non-fluorescent particles are counted. It should be noted that due to the limitation of xenon flashlamp recharge time, some particles which meet the trigger conditions may not be interrogated by flashlamp during the sampling procedure and only their particle sizes and asphericity factors are recorded. These non-excited particles are picked randomly during the sampling procedure regardless of their properties. When calculating the number concentrations (marked as $N_{xx}$, 'xx' refers to an aerosol type of aerosols) of excited particles (including fluorescent particles), compensation will be factored in by multiplying the ratio of the concentration of total particles to the concentration of excited particles.

To better understand the sources of different types of aerosol particles and minimize the effect of atmospheric boundary layer development, the number fractions to total particles of each type of aerosol particle (i.e. $N_{xx}/N_{Total}$, marked as $F_{Total}(xx)$) and the number fractions to whole fluorescent particles of each type for fluorescent aerosol particles (i.e. $N_{xx}/N_{Fluor}$, marked as $F_{Fluor}(xx)$) are investigated. During observation, the variations of each type of aerosol particle in their size and asphericity factor distribution can also predict some atmospheric events such as hygroscopic growth and dust transport. In this paper, the count mean diameters (marked as $Mean\ D_{xx}$) and count mean asphericity factors (marked as $Mean\ AF_{xx}$) of each investigated type of aerosol particle are used to present their variations in their size distribution and AF distribution. Detailed definitions of the above abbreviations and parameters are listed in Appendix A.

During observation, a WIBS instrument (NEO model) is located in a room on the top floor of the SESS building. within 30 m away from those meteorological instruments mentioned before. The atmospheric aerosols are sampled by WIBS through a sampling tube extending about 2 m outside the building. The FT process is performed every 8 hours during the observation and the thresholds at a specific time are determined by the linear interpolation of two adjacent FT procedure results. Although the lower limit for particle size measurement is 0.5 μm, only particles with a size larger than 0.8 μm are discussed in this paper for consistency with other studies and excluding potential interferents. The following ten types of aerosol particles are counted every 30 min: total particles, non-fluorescent particles, seven types of fluorescent particles, and whole fluorescent particles.

### 2.4. Backward trajectory analysis using the HYSPLIT model

To identify aerosol sources and the transport path, the HYSPLIT model (Stein et al., 2015) is used in this study. The HYSPLIT model is configured to use meteorological data from the Global Data Assimilation System (GDAS) at a spatial resolution of 1° for performing a 48 h backward trajectory computation. Map data from Natural Earth (naturalearthdata.com) are used to show the HYSPLIT backward trajectory result.

## 3. Result and discussion

### 3.1. Aerosol transport event on 13 March

#### 3.1.1. Lidar and in situ observation

Figure 1(a–e) shows the time-height cross-section of the attenuated backscatter coefficient, Doppler spectral width, and wind vector over Hefei. The difference in wind direction between wind at high altitudes and near the surface can be seen at 0:00. With the downward development of wind at high altitudes, the wind near the surface changes direction from south to north at about 3:30. At the same time, an external aerosol layer occurs at a height of 1.5 km accompanied by high-speed horizontal wind (~10 m s$^{-1}$). When the external aerosol layer reaches the ground along with the high-speed wind at 7:30, the attenuated backscatter coefficient near the surface shows enhancement, and significant increases in local PM concentrations can be observed (Fig. 1(f)) in the meantime. After 9:00, the detection range of lidar drops sharply to about 500 m due to strong optical attenuation near the surface. The Lidar signals at the largest detection range show stronger backscatter and much broader Doppler spectral width compared to that near the surface layer during this period. The above phenomena indicate that a low-altitude cloud layer with a high concentration of aerosol moves to Hefei after 9:00.

It should be noted that PM$_{10}$ (Fig. 1(f)) reaches its maximum concentration of 122 μg m$^{-3}$ and starts to decrease at 10:00, 1 hour after the low-altitude cloud layer is observed by the lidar system. In contrast to PM$_{10}$, the PM$_{2.5}$ concentration sharply increases after 10:00 and reaches a maximum of 110 μg m$^{-3}$ at 12:00, which is also the highest PM$_{2.5}$ concentration record between 11–20 March. When the cloud layer stays at the low-altitude layer between 9:00 and 21:00, local weather conditions (Fig. 1(i)) show high humidity (78 %–89 %) and low temperatures (6 °C–13 °C), which inhibit aerosol diffusion but favor the accumulation of local aerosols and hygroscopic growth of particles. As such, the increase in particulate matter concentration from 7:00 to 9:00 is mainly attributed to external aerosol transport. The different trend between PM$_{2.5}$ and PM$_{10}$ concentrations after 10:00 can be attributed to their different removal efficiency: large particles have higher hygroscopic growth factors (Haenel et al., 1978; Hänel, 1976), their sizes increase largely through hygroscopic growth, and thus they have a higher probability to be removed by gravitational settlement than smaller particles. As for PM$_{2.5}$, they have lower removal efficiency and their concentration can continue to increase through the intensifying local anthropogenic aerosol emission after morning and the inhibited aerosol diffusion. During the period, PM$_{2.5}$ is observed to exceed the PM$_{10}$ concentration at one time, which can be attributed to the difference between instruments and methods for monitoring PM$_{2.5}$ and PM$_{10}$. After 12:00, the horizontal wind near the surface accelerates and a precipitation event occurs from 15:00–18:00 (Fig. 1(j)). These two factors contribute to local aerosol diffusion and removal and explain the decrease in local PM concentrations after 12:00. WIBS data (Fig. 1(g)) show that local aerosol number concentration significantly increases from 8:00, and reaches its maximum number concentration of 11.93 cm$^{-3}$ at 10:00, which is the highest number concentration observed by WIBS between 11–20 March. The size distribution variation (Fig. 1(h)) reveals the greatest increase in aerosol particles comes from sub-micron particles. The different behavior of PM data and WIBS data may be due to the difference in observation location and monitoring

method (As mentioned in Sec.2.2., the nearest PM monitoring location is about 2.7 km northwest of the USTC campus).

### 3.1.2. Categorized WIBS data

WIBS statistics on 13 March are shown in Fig. 2. $N_B$, $N_C$ and $N_{BC}$ sharply increase from 8:00, 30 minutes after the high-speed wind at high altitude reaches the ground, and finally reach their peak at about 10:00, which is consistent with the $N_{Total}$, $N_{NonFluor}$, $N_{Fluor}$ and the PM$_{10}$ concentration (Fig. 1 (f)). Accompanied with their increasing number concentration, these types of aerosol particles except Type BC exhibit a sharp decrease in their $Mean\ D$ and $Mean\ AF$, and reach their daily minimum $Mean\ D$ during this period. However, Type AB and Type ABC particles have different time variation trends from the above-mentioned fluorescent particles: $N_{AB}$ and $N_{ABC}$ both start to increase at about 3:30 when the wind near the surface changes (Fig. 2(b)) and they reach their peak at about 7:00–8:00, causing $F_{Fluor}(AB)$ and $F_{Fluor}(ABC)$ to increase from 1.6 % and 7.2 % to their maximum of 6.5 % and 16.9 % respectively, and consequently leading $F_{Total}(Fluor)$ to increase to its daily maximum of 34.7 % at 6:30. Meanwhile $Mean\ D_{AB}$ and $Mean\ D_{ABC}$ reach their daily minimum but no obvious trends are observed in their $Mean\ AF$. Although $N_A$ shows a similar trend like Type B, Type C, and Type BC particles, and reaches its peak at 10:00, an increase of $F_{Total}(A)$ can be seen after 3:30, as $N_{AB}$ and $N_{ABC}$ start to increase, and reaches its daily peak at 8:00, a minor peak of $F_{Fluor}(A)$ can also be seen during this period. These phenomena indicate the growth rate of $N_A$ between 3:30–7:30 is higher than that of $N_{Total}$. Type BC particles reach their maximum concentration at 10:00 but $Mean\ D_{BC}$ decreases only very slightly at this time, reaching the daily minimum value at 16:00 during the rainfall event and a second minimum value at 5:30 when $N_{AB}$ and $N_{ABC}$ increase. Moreover, Type AC particles show an obscure time variation due to their extremely low concentration and are excluded from further discussion.

The differences in these types of fluorescent aerosols reveal that the increased aerosols between 7:30–10:00 have different sources from those between 3:30–7:30. Judging from their number concentrations and fractions mentioned before, Type A, Type AB, ABC aerosols reach Hefei from the north after 3:30 when the wind near the surface changes its direction, among which the Type AB and Type ABC are dominant. There may also be a tiny fraction of Type BC particles judged from decreased $Mean\ D_{BC}$. The majority of these are in the fine mode so their mean diameters decrease during this period. After 7:30, as the external aerosols at high altitudes are transported to the ground with high-speed winds, most types of aerosols have not only increased concentrations but also decreased $Mean\ D$ and $Mean\ AF$. Considering the high humidity (> 80 %) and low temperature on that day which favor hygroscopic growth, their decreased $Mean\ D$ and $Mean\ AF$ can be explained by the hygroscopic growth of transported particles whose origin sizes are below the detection range. Due to hygroscopic growth or aggregation and the resulting deposition of large-size particles, Type AB and ABC aerosols have increased $Mean\ D$ but decreased $N$ during this period. After 12:00, all types of aerosol particles decrease sharply in concentration as the horizontal wind accelerates.

Moreover, $N_A$, $N_{AB}$, and $N_{ABC}$ and $F_{Fluor}$ begin to increase after a drizzle event observed at 15:00 (Fig. 2(i)). This phenomenon is similar to the previous observation in Beijing (Yue et al., 2016) and can be explained by the wet discharge after rainfall.

### 3.1.3.    Transport path and transported bioaerosol types

HYSPLIT backward trajectory results (Fig. 3) show the difference in direction between winds near the surface and at high altitudes. The wind near the surface has a southerly direction between 0:00 and 4:00. However, the wind at high altitude has a different direction when traveling over Hefei and changes direction earlier than the wind near the surface. At 4:00, the wind direction at high altitude changes from the southeast to the north while the wind on the surface is still southerly. At 8:00, the local wind near the surface changes to a northerly direction that is consistent with the wind at high altitudes. At 12:00, when the $PM_{2.5}$ concentration in Hefei reaches its maximum, both the wind near the surface and at high altitude show a transport path passing several coastal cities in Yangtze River Delta Urban Agglomerations such as Hangzhou, Ningbo, and Shanghai, which can be a source of wet air mass and pollutants. The transported pollutants, the increasing emission of local anthropogenic aerosol after sunrise, the accumulation of aerosols caused by the low-altitude cloud layer, and the hygroscopic growth of small size aerosol particles under high humidity altogether contribute to the rapid increase of PM concentration. These increased aerosol particles have a less fraction of fluorescent biological aerosols, which explains the rapid decrease of fluorescent particles in their fraction to total particles observed by WIBS after 7:30.

The increased aerosol particles between 3:30–7:30 are more worth noting during this event because they have typical fluorescent characterizations of bioaerosol. As discussed before, the increased aerosol particles during this period contain Type A, Type AB and Type ABC particles, among which the Type AB and Type ABC are dominant, and their decreased *Mean D* during this period (Fig.2 (d)(g)(j) in the right panel) show that their sizes are mainly distributed in the range of less than 1.5 μm. Previous works (Hernandez et al., 2016; Savage et al., 2017) use WIBS to characterize multiple classes of fluorescent biological aerosol particles and non-biological fluorescent interferent samples in a laboratory setting. A summary of the results of these works is listed in Appendix B for comparison with our works. Detailed information can be found in these references. Compared with laboratory tests, the increased fluorescent aerosol particles have a typical size of bacteria (<1.5 μm) but their fluorescent characterization shows a mixture of Type A, AB, and ABC, which are typical fluorescent types of fungal spores. Pollen fragments are excluded because their typical sizes are much larger and their typical fluorescent types are Type C, BC, and ABC, which is different from that in this event. Some kinds of spore-forming bacteria such as *Bacillus subtilis* shows higher fluorescent intensity than other bacteria and are mainly categorized as Type AB particles in a laboratory test (Hernandez et al., 2016) and their larger size multicell aggregates in atmospheric might be categorized as Type ABC because larger fluorescent particles intend to have higher fluorescent intensities (Savage et al., 2017; Yue et al., 2019). Fungal spores can also be found in submicron aerosol particles (Xu et al., 2017), indicating that some of them can have smaller sizes than the laboratory test result. So, it can be inferred from the characterization of increased fluorescent aerosol particles that these transported external bioaerosols are most likely to be bacteria aggregates or fungal spores. Backward trajectory results indicate their sources can be the nearby rural area north of Hefei.

### 3.2. Aerosol transport event on 16–17 March

#### 3.2.1. Lidar and in situ observation

Figure 4(a–e) shows the lidar observation results during 16–17 March. A developing cloud layer that extends downwards can be seen on 16 March. This cloud layer reaches its lowest height of 2 km at 22:00 on 16 March before moving upward and disappearing. During this period, a high-speed horizontal wind is observed in the upper cloud layer with a high vertical speed (~2 m s$^{-1}$). A broadened Doppler spectral width is also observed underneath the cloud. In previous works, a fall velocity greater than 1 m s$^{-1}$ can be identified as precipitation (Manninen et al., 2018; Wang et al., 2019a) and Doppler spectral will be broadened during precipitation due to additional signal peaks from raindrops (Wei et al., 2021). Although these characteristics are the same as a precipitation event, no rainfall is recorded on the ground during this event (Fig.4(j)). These observations indicate a virga event, during which the hydrometeors beneath the cloud layer enhance lidar backscatter and evaporate before reaching the ground. During the virga event, the wind at high altitude migrates downward, causing the southeast wind near the surface to weaken and finally change to the northwest direction at 01:30 on 17 March, making it consistent with the wind at high altitude. Meanwhile, an external aerosol layer can be observed at a height of about 0.5–1.5 km, which may be from the accumulation of aerosol that originally acted as ice nuclei during the virga event. Due to the mixing of the local aerosol and external aerosol carried by the wind at high altitude and the increase in PM$_{2.5}$ and PM$_{10}$ concentration (Fig. 4(f)), the total particle number concentration of WIBS (Fig. 4(g)) is observed. The ground observation result shows a maximum PM$_{2.5}$ concentration (Fig. 4(f)) of 71 μg m$^{-3}$ at 10:00 on 17 March, which is the second-highest peak concentration between 11–20 March. WIBS data show a higher fraction of coarse particles at this time than that during the event on 13 March. Under the influence of solar radiation, local temperature and relative humidity show a sharp increase and decrease respectively (Fig. 4(i)). The Doppler spectral width profile (Fig. 4(b)) shows that compared to values on 16 March, the convective boundary layer on 17 March has a much higher maximum height of more than 2 km at about 15:30 which favors the diffusion of aerosol. The aerosol concentrations decrease sharply under strong diffusion after 10:00 on 17 March.

#### 3.2.2. Categorized WIBS data

The statistical results of WIBS data between 16–17 March are shown in Fig. 5. The number concentration of each type of aerosol increases during the transport event (Fig. 4). Their maximum concentrations are all observed at about 8:30, which is consistent with the time of maximum PM$_{10}$ concentration (Fig. 4(f)). While number concentrations increase, different types of fluorescent aerosol particles show different trends in their number fraction to whole fluorescent aerosol particles. For example, $F_{Fluor}(BC)$ shows a significant drop from 48.4 % at 6:00 to 40.9 % at 8:30 (Fig. 5 (i)) but $F_{Fluor}(A)$, $F_{Fluor}(AB)$ and $F_{Fluor}(ABC)$ increase from 1.8 %, 1.6 % and 3.7 % at 6:00 to 3.8 %, 3.7 % and 9.3 % at 8:30 on 17 March respectively (Fig. 5 (d)(g) and (j)). These changes that occur in a short amount of time reveal that the transport of aerosols not only leads to high PM$_{10}$ and PM$_{2.5}$ concentrations but also leads to the increase of some types of fluorescent particles in their fraction to whole fluorescent particles.

The variation in mean diameters (Fig. 5) reveals that Type B, C, and BC particles share the same variation trend with non-fluorescent and total particles. Their $Mean\ D$ keep decreasing when their $N$ increase slightly between 1:30 and 6:00 on 17 March, followed by increases in both their $Mean\ D$ and $N$ until 8:30. The phenomenon indicates that the increase in $N$ consists of two stages. From 1:30 to 6:00, the increased number of aerosol particles are mainly in fine mode; but from 6:00 to 8:30, the increased number of particles have higher fractions of coarse mode. However, this two-stage variation is not observed in all Type FL-1 particles (Type A, AB, AC, ABC). The Type FL-1 particles do not show increases in number concentrations during stage 1 but show sharp increases in stage 2. And their $Mean\ D$ show no apparent variation (Fig. 5 (d) (g) (h) and (j)) during the 2 stages. It can be explained that from 1:30 to 6:00 on 17 March the increase in particle concentrations mainly resulted from the local aerosol accumulation and possible hygroscopic growth caused by rising humidity and low turbulence intensity (Fig. 4(b)). After 6:00 on 17 March, turbulence intensity begin to increase as sunrise occurred, which inhibit local aerosol accumulation increased particles are dominated by transported external aerosols in which there are a higher fraction of particles in the coarse mode. The sharp increase in $N$ of Type FL-1 particles indicates their main source is the transported external aerosols.

### 3.2.3. Transport path and transported bioaerosol types

The backward trajectory results shown in Fig. 6 reveal that during the transport event, the wind near the surface changes direction in Hefei from the southeast at 22:00 on 16 March to the west at 6:00 on 17 March, which is consistent with lidar observation shown in Fig. 4(d). The difference in direction between wind near the surface and at high altitude over Hefei at 10:00 on 17 March is supported by lidar observation (Fig. 4 (d)) and can be explained by local turbulence after sunrise. Trajectory results show that the air mass at both levels passes through the Dabie Mountains west of Hefei which can be a source of bioaerosols. The sharp increases of Type FL-1 particles between 6:00–8:30 are worth noting here because they have typical characteristics of bioaerosol. According to the variation of their $Mean\ D$ during this period, the sizes of increased Type FL-1 particles are distributed in the range of 1.3 μm–1.8 μm and their fluorescent types show a mixture of Type A, AB, ABC and a tiny fraction of AC particles. According to previous laboratory tests (Hernandez et al., 2016; Savage et al., 2017) and previous discussion in the 13 March event, these characteristics are similar to that of fungal spores. However, bacteria aggregates cannot be excluded in this event, because in the atmospheric environment, bacteria aggregates can be confused with fungal spores because of their similar sizes and fluorescent spectra (Hernandez et al., 2016) .increased fluorescent particles can both be fungal spores or bacteria aggregates. Backward trajectory result shows that their sources can be the Dabie Mountains in the west of Hefei.

### 3.3. Aerosol transport event on 19 March

### 3.3.1. Lidar and in situ observation

As portrayed in Fig. 7 (a–e), an external aerosol layer accompanies the high-speed northwest wind (~20 m s$^{-1}$) and is observed at a height of 2–3 km at 2:00. It moves down to the ground at 5:00. Meanwhile, the wind near the surface changes direction from southwest to northeast, which is consistent with the wind at the high altitude. Strong wind shear at this moment broadens the Doppler spectral width shown

in Fig. 7(b). After 5:00, with the development of convection, the thermal mixing layer is elevated, which favors surface aerosol diffusion. An entrainment layer with relatively low wind speed and strong backscatter occurred above the mixing layer after the transport of external aerosol and moved upward to above 2 km at 12:00. After 12:00, interference between the entrainment layer and the thermal mixing layer can be observed. $PM_{10}$ concentrations (Fig. 7 (f)) show a significant increase from 111 μg m$^{-3}$ at 7:00 to 524 μg m$^{-3}$ at 10:00 with the enhancement of attenuated backscatter coefficient near the surface. This is the highest $PM_{10}$ record between 11–20 March, but no obvious increase in $PM_{2.5}$ concentration is observed. WIBS data (Fig. 7 (g) (h)) show that particles in coarse mode are most abundant during this period. Temperature is rising and humidity decreasing (Fig 7. (i)) while $PM_{10}$ is increasing, which inhibits hygroscopic growth and accumulation of aerosol particles. So, it can be inferred that the sharp increase of $PM_{10}$ concentration cannot be attributed to the local aerosol accumulation or hygroscopic growth, but the transport of external aerosols. After sunrise, with an increase in solar radiation, the PBL height rises, and aerosol diffusion increases. The increased aerosol diffusion contributes to the decrease of PM concentration after 10:00.

### 3.3.2. Categorized WIBS data

Figure 8 reveals that as the external aerosol reaches the ground (Fig. 7(a)), $Mean\ D$ and $Mean\ AF$ of all types of fluorescent aerosol particles increase sharply at different degrees from 6:00 on 19 March, which is higher than that in the two events described before and also the highest record during the observation period. The impact from the external aerosol indicates that the external aerosol layer is dominated by non-spherical aerosol particles in coarse mode. Although all other types of fluorescent particles exhibit an increase in their number concentration during this event, Type BC shows a different trend. As portrayed in Fig. 8(i), $N_{BC}$ abnormally decreases from 0.68 cm$^{-3}$ at 6:00 to 0.52 cm$^{-3}$ at 9:00, causing $F_{Fluor}\ (BC)$ to decrease from 57.6 % at 6:00 to 36.4 % at 9:00 and $F_{Total}\ (BC)$ to decrease from 25.8 % at 6:00 to 10.6 % at 9:00. Considering its high concentration before 6:00, the sharp decline of $N_{BC}$ also lead to the rapid decrease of $F_{Total}\ (Fluor)$, which decrease from 44.7 % at 6:00 to 29.2 % at 9:00 (Fig. 8(b)). To compensate for the decreased $N_{BC}$, all other types of fluorescent particles show increases in their $F_{Fluor}$ in different degrees, which is predictable. However, when considering the $F_{Total}$ of other types of fluorescent particles to exclude the influence of decreased $N_{BC}$, they do not exhibit sharp decreases in $F_{Total}$ like Type BC particles does, only $F_{Total}\ (C)$ shows slightly decrease from 6.6 % to 5.3 % (Fig. 8(f)). On the contrary $F_{Total}\ (A)$ and $F_{Total}\ (AB)$ increase from 0.7 % and 0.5 % at 6:00 to 1.4 % and 0.9 % (Fig. 8 (d) and (g)). From the almost the similar decline range of $F_{Total}\ (BC)$ and $F_{Total}\ (Fluor)$ and no similar sharp decrease of $F_{Total}$ observed in other types of fluorescent particles, it can be concluded that the major reason for the significant drop of $F_{Total}\ (Fluor)$ are the abnormal decrease of $N_{BC}$ during the event. In addition, the increase of $F_{Total}\ (A)$ and $F_{Total}\ (AB)$ during transport indicates that the external transported aerosol has a higher fraction of Type A and Type AB particles and a much lower fraction of Type BC particles than local aerosols.

### 3.3.3. Transport path and transported bioaerosol types

The backward trajectory result on 19 March (Fig. 9) shows the wind direction gradually changing near the surface and at high altitudes over Hefei. From 6:00, the direction of the wind near the surface changes from southwest to northeast, which is consistent with the wind at high altitudes. At 8:00, the high-altitude airmass over Hefei shows a new transport path. As the wind at high altitude migrates downwards, the airmass at both heights over Hefei at 10:00 shows similar transport paths. The long-range transport path indicates a high-speed northerly wind, which is consistent with the lidar observation result. The transport path at both heights passes the Gobi Desert at the China-Mongolia border, which is one of the main sources of dust storms in East Asia.

The increased mean diameter and asphericity factor of all types of aerosol particles, the transport path, and the high $PM_{10}$ concentration during the event altogether indicate the transported external aerosols are dominant by large-size non-spherical particles, which are most likely to be mineral dust. Although $F_{Total}(Fluor)$ decreases during the transport event, the increasing $F_{Total}(A)$ and $F_{Total}(AB)$ during the transport event still indicates the potential bioaerosol transport, because bacteria and fungal spores are related with Type FL-1 (usually Type A, AB, ABC particles) particles (Hernandez et al., 2016; Savage et al., 2017). According to their mean diameter, the sizes of these transported Type A and Type AB particles during this event are mainly distributed in the range of larger than 1.8 μm for Type A particles and larger than 2.0 μm for Type AB particles, which means the potential transported bioaerosol particles are the largest among the three transport events. However, it is not appropriate to assume that these increased Type A and Type AB are individual bioaerosol particles. Previous works (Savage et al., 2017; Yue et al., 2019) show that larger fluorescent biological aerosol particles are more likely to exhibit a higher fluorescent intensity and wider fluorescent spectrum range, which means exhibit fluorescent in multiple fluorescent channels in WIBS measurement. For example, larger size fungal spores and pollen have higher probabilities to be categorized as Type ABC particles than smaller bacteria. During this event, the transported external aerosol particles have larger sizes than local aerosol particles, however, the expected particles with the highest fluorescent intensities, Type ABC particles do not show an obvious increase in their fraction to total particles. These phenomena indicate that the larger size transported external fluorescent particles do not have an apparently higher fraction of Type ABC particles, and thus have lower fluorescent efficiency than local fluorescent particles. In addition, considering the transported external Type A and Type AB particles have the particle size and asphericity factor characteristics like mineral dust, it is a better explanation that these transported external large-size Type A and Type AB particles result from the bacteria that attached to dust. Researches reveal that microbial activity is significant in the aerosols from desert regions, even impacting the composition of aerosols in downwind regions (Ho et al., 2005; Hua et al., 2007; Maki et al., 2014, 2015; Tang et al., 2018). During long-range transport, larger mineral particles attached by bacteria can serve as a shelter and favor the survival of bacteria. Dust-attached bacteria have been found in SEM (scanning electron microscope) images from air samples of previous research (Tang et al., 2018). A field campaign (Maki et al., 2019) in the Gobi Desert, which is the source of this event, reveals that after dust events, bacteria from Bacteroidetes, which are known capable of attaching to coarse particles, increase their relative abundance in air samples. The above results support our explanation of WIBS data. The pathogenic bioaerosol

during dust transport events is believed to be linked to allergen burden and asthma (Ichinose et al., 2005; Liu et al., 2014), and even multiple diseases such as Kawasaki disease in humans (Rodó et al., 2011) and rust diseases in plants (Fröhlich-Nowoisky et al., 2016). Dust transport events generally occur during spring in Hefei. The WIBS data during this event indicates that the long-range dust transport during spring has potential risks to human health in the Hefei area.

In addition, the abnormal decrease in the concentration of Type BC particles should also be noted during this event. Laboratory test (Hernandez et al., 2016) shows that Type BC particles are not a typical fluorescent type for bacteria and fungal spores. Bacteria and fungal spores are mainly connected with Type FL-1 particles. On contrary, EEM (excitation-emission matrix) analysis result from sampled aerosol (Chen et al., 2016, 2020; Wen et al., 2021; Yue et al., 2019) shows that highly-oxygenated HULIS (humic-like substances) has a similar fluorescent fingerprint to Type BC particles and can thus be non-biological fluorescent interferent during WIBS measurement by categorized into Type BC particles. This point is supported in an Mt. Tai field campaign (Yue et al., 2019) by the moderate correlation between Type BC particles and highly-oxygenated HULIS. Atmospheric HULIS has many sources including primary emission mechanisms like biomass burning, and multiple secondary formation processes (Wu et al., 2021). Compared with rural areas, there are abundant air pollutants of strong oxidation capacity in urban areas like NOx and O3, which may contribute to the formation of highly-oxygenated organic aerosol like HULIS (Tong et al., 2019). This can explain the high concentration and fraction of local Type BC particles and the abnormal decrease of its concentration and fraction during external aerosol transport. This explanation is adopted by previous research (Wen et al., 2021), where they also find the decrease of highly-oxygenated HULIS concentration in the urban area during the dust period.

## 4. Conclusion

In this paper, three aerosol transport events between 11–20 March 2020 in Hefei were investigated. The results of a coherent Doppler wind lidar and a bioaerosol sensor WIBS, local meteorological parameters, local PM concentrations, and the HYSPLIT backward trajectory model were used during the investigation. The analysis result was summarized as follows.

During all three transport events, differences between the directions for wind at high altitude and near the surface and the downward migration of wind at high altitude were observed. Enhancement in the attenuated backscatter coefficient near the surface was captured as the wind at high altitude reached the ground, and a corresponding surface PM concentration increase was monitored during the events. During the observation, the 13 March and 16–17 March events contributed to the highest and second-highest $PM_{2.5}$ concentration record respectively, and $PM_{10}$ concentration reached the highest record during the 19 March event.

During the above three transport events, seven types of fluorescent particles, as well as non-fluorescent particles, were categorized and counted using WIBS data. The fractions to total particles and the fractions to fluorescent particles for Type AB and Type ABC particles increased during the 13 March and 16–17 March events. In the 19 March event, the fraction to total particles of Type A and Type AB particles increased, while all other types of fluorescent particles showed a decreasing trend. These phenomena proved the influence of transported external bioaerosols. The possible origins of external

bioaerosols are discussed according to WIBS statistics and trajectory results: In the 13 March event, they are fungi or bacteria aggregates from rural areas in the north of Hefei; in the 16–17 March event, they are fungi or bacteria aggregates from Dabie Mountains in the west of Hefei; in the 19 March event, they are dust-attached bacteria originating from Gobi Desert in China-Mongolia border.

The combination of UV-LIF based online measurement instruments and ground-based coherent Doppler wind lidar expands the aerosol monitoring parameters and proves to be a potential method for real-time monitoring of fluorescent biological aerosol transport events. By utilizing this method, this paper not only observed the impact of external bioaerosol transport on local aerosol composition, but also discovered the relationships between the atmospheric processes at high altitudes like virga and variation of surface bioaerosol. It contributes to the further understanding of long-range bioaerosol transport, the roles of bioaerosols in atmospheric processes and in aerosol-cloud-precipitation interactions.

## Appendix A

**Table A1. Description of abbreviations and parameters in the text**

| Abbreviation | Description |
|---|---|
| Total | Total aerosol particles detected by WIBS |
| Fluor | Fluorescent aerosol particles detected in any one of the three channels |
| Type FL-1 | Fluorescent aerosol particles detected in channel FL1 |
| Type FL-2 | Fluorescent aerosol particles detected in channel FL2 |
| Type FL-3 | Fluorescent aerosol particles detected in channel FL3 |
| Type A | Fluorescent aerosol particles detected in channel FL1 only |
| Type B | Fluorescent aerosol particles detected in channel FL2 only |
| Type C | Fluorescent aerosol particles detected in channel FL3 only |
| Type AB | Fluorescent aerosol particles detected in channels FL1 and FL2 |
| Type AC | Fluorescent aerosol particles detected in channels FL1 and FL3 |
| Type BC | Fluorescent aerosol particles detected in channels FL2 and FL3 |
| Type ABC | Fluorescent aerosol particles detected in channels FL1, FL2, and FL3 |
| $N_{xx}$ | The number concentration of a certain type of aerosol particles |
| $F_{Total}$ (xx) | Number fraction to total aerosol particles of a certain type of aerosol particles, i.e. $N_{xx}/N_{Total}$ |
| $F_{Fluor}$ (xx) | Number fraction to fluorescent aerosol particles of a certain type of aerosol particles, i.e. $N_{xx}/N_{Fluor}$ |
| Mean $D_{xx}$ | Count mean particle diameter of a certain type of aerosol particles detected by WIBS |
| Mean $AF_{xx}$ | Count mean asphericity factor of a certain type of aerosol particles detected by WIBS |

## Appendix B

The following table presents typical characterizations of different types of fluorescent biological aerosol particles and potential non-biological fluorescent interferents detected by WIBS. All following results are retrieved from laboratory tests using the old model WIBS-4 instead of WIBS-NEO used in this paper. However, these results should not be treated as absolute "signatures" but references for discrimination between broad bioaerosol categories, due to the differences in instrumental parameters, culture circumstances, and measuring environment. For example, bacteria often exist in the form of multi-cell agglomerates or particle mixtures in the atmospheric environment, exhibiting large sizes than a single cell observed here (Hernandez et al., 2016). Besides, pollens have wider size distribution than fungi due

to the fragmentation of larger intact pollen grains, some of which have sizes larger than 10 μm and even beyond the upper limit of WIBS-4 detection (~20 μm) (Hernandez et al., 2016; Savage et al., 2017). And finally, compared with the old WIBS-4, the new model WIBS-NEO maintains the same fluorescence

560    channel configuration but upgrades the data acquisition module of size detection and fluorescence intensity measurement, which may also introduce potential variabilities from these laboratory results.

**Table B1. Typical aerosol characterization in WIBS detection**

| Species | Diameter | AF | Most frequent Types in WIBS | References |
|---|---|---|---|---|
| Bacteria | <1.5 μm | | A | Hernandez et al., 2016 |
| Fungal Spores | 2–9 μm | | A, AB, ABC | |
| Pollen (Fragments) | 2–9 μm | | C, BC, ABC | |
| Bacteria: *Bacillus atrophaeus* | 2.2±0.4 μm | 17.4±4.1 | A | Savage et al., 2017 |
| Bacteria: *Pseudomonas stutzer* | 1.1±0.3 μm | 19.2±2.8 | A | |
| Fungal Spores: *Aspergillus brasiliensis* | 3.6±1.8 μm | 20.8±10.3 | A, AB | |
| Fungal Spores: *Saccharomyces cerevisiae* | 7.2±3.7 μm | 28.7±16.8 | AB, ABC, A | |
| Pollen Fragment: *Phleum pratense* | 6.0±3.2 μm | 23.1±13.4 | A, AB, ABC | |
| Pollen Fragment: *Alnus glutinosa* | 6.1±3.2 μm | 25.2±14.6 | B, AB, BC, ABC | |
| Dust: Gypsum | 4.1±3.0 μm | 19.3±12.2 | Non-Fluor | |
| HULIS: Suwannee River fulvic acid standard I | 1.7±1.0 μm | 12.0±10.1 | Non-Fluor, B | |
| Brown Carbon: Glycolaldehyde + methylamine | 1.2±0.4 μm | 17.9±24 | Non-Fluor, B | |
| Brown Carbon: Glyoxal + ammonium sulfate | 1.3±0.6 μm | 14.1±3.5 | A, B, AB | |
| Soot: Diesel soot | 1.1±0.4 μm | 21.2±10.1 | A | |
| Soot: Wood smoke (Pinus Nigra) | 1.0±0.7 μm | 9.5±4.3 | Non-Fluor, B, AB | |

**Data availability.** The GDAS data in HYSPLIT backward trajectory are publicly available from the

565    NOAA website at https://www.ready.noaa.gov/hypub-bin/trajasrc.pl. All other data in this study are available from the authors upon request.

**Author contributions.** JY performed the lidar observations. TW downloaded the local meteorological data and derived the parameters needed from the raw lidar data. DT performed the WIBS observations, HYSPLIT backward trajectories, carried out the data analysis, prepared the figures, and wrote the original

570    draft. All authors contributed to the interpretation of discussion data, manuscript reviewing, and editing.

**Competing interests.** The authors declare that they have no conflict of interest.

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

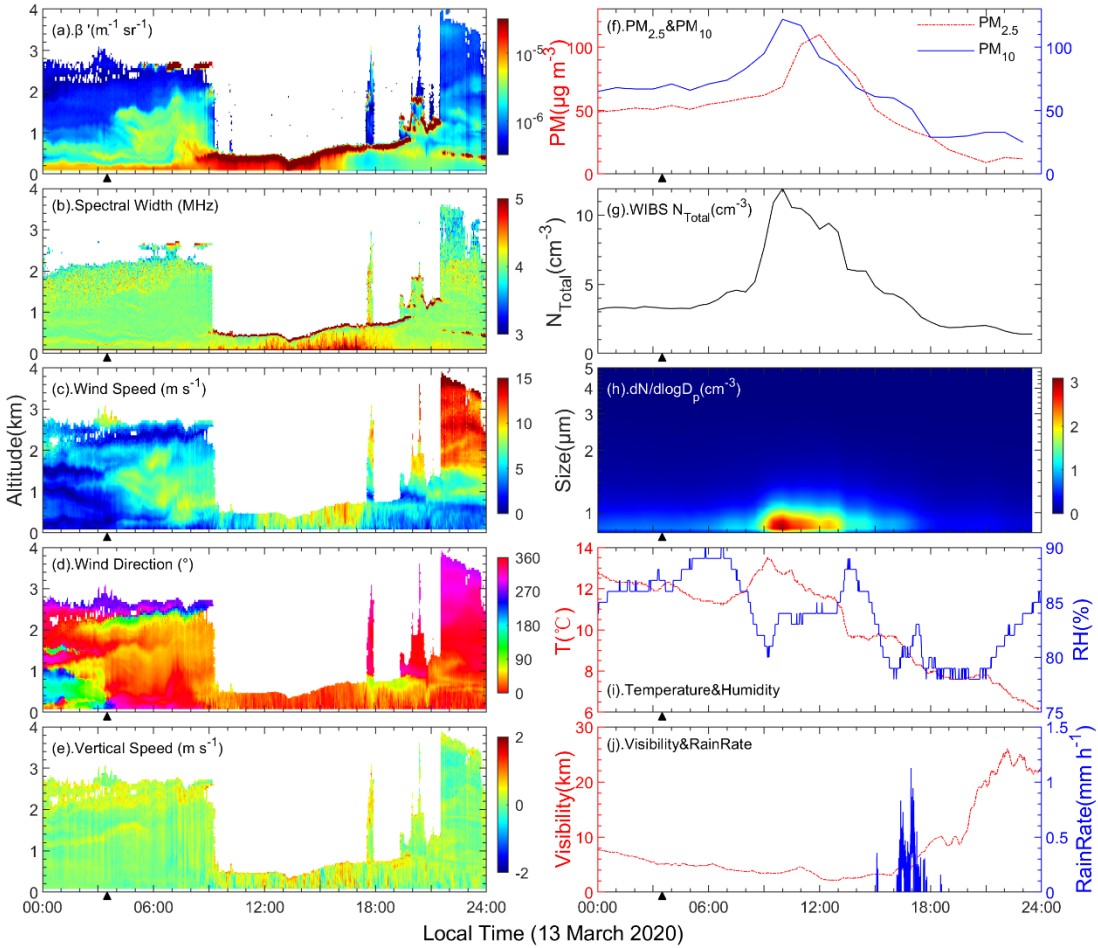

**Fig. 1 Left panel: Time-height crosssection of (a) attenuated backscatter coefficient, (b) Doppler spectral width, (c) horizontal wind speed, (d) horizontal wind direction, and (e) vertical wind speed over Hefei observed by CDWL on 13 March 2020. Right panel: simultaneous ground observation results, including (f) surface hourly particulate matter concentration from local monitor network, (g) number concentration and (h) size distribution of surface total aerosol particles from WIBS, (i) local temperature and humidity, and (j) local visibility and rain rate. The wind direction near the surface changes at about 03:30 and is marked with a triangle symbol.**

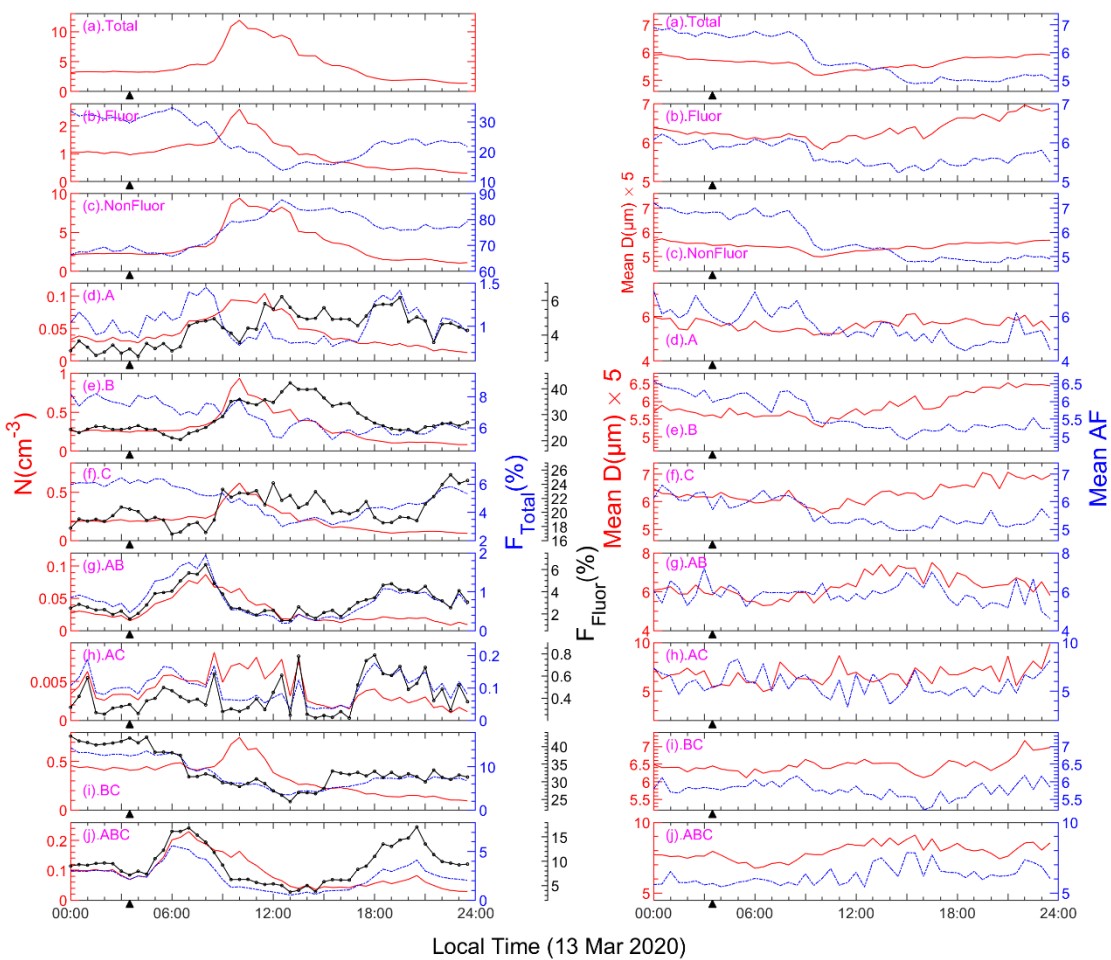

Fig. 2 Left panel: number concentrations (solid red line), number fractions to total particles (blue chain line), and number fractions to fluorescent particles (black solid line with point marker) of the investigated particle types. Right panel: Count mean particle diameter (×5, solid red line) and count mean asphericity factor (blue chain line) of investigated particles measured by WIBS on 13 March 2020. The direction of the wind near the surface changes at about 03:30 and is marked with a triangle symbol.

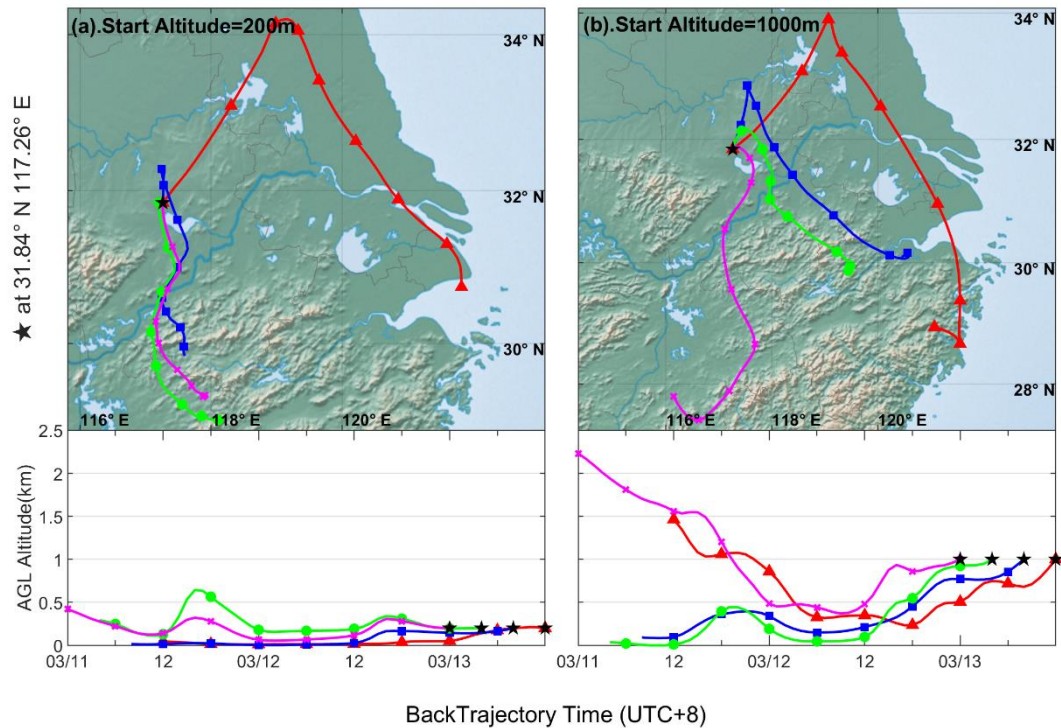


**Fig. 3 The 48 h HYSPLIT backward trajectory result calculated at 0:00 (magenta line with cross marker), 4:00 (green line with diamond marker), 8:00 (blue line with square marker), and 12:00 (red line with triangle marker) on 13 March 2020 (UTC+8). Terminal points are marked every 6 hours. The starting location is set as 200 m (left panel) and 1000 m (right panel) over Hefei. Made with Natural Earth. Free vector and raster map data @ naturalearthdata.com.**


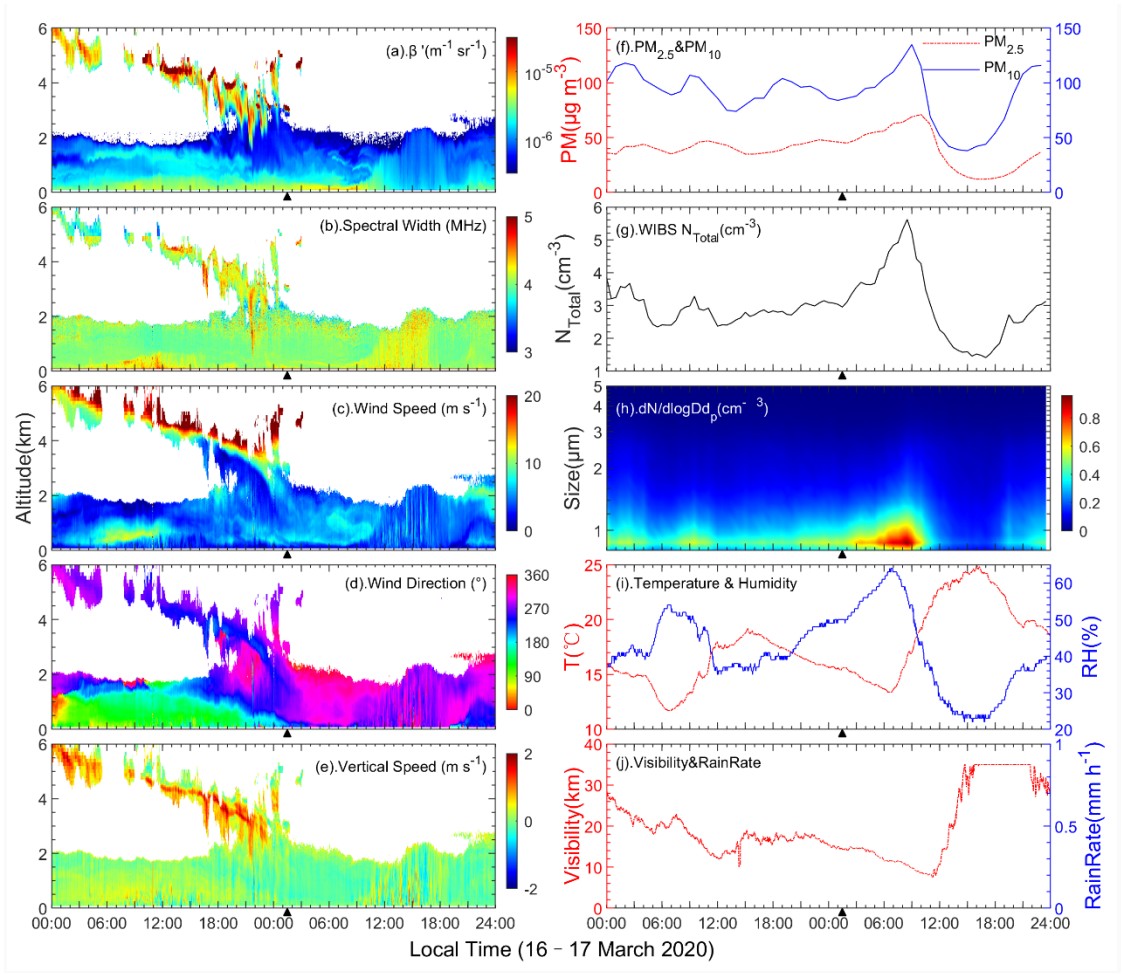

**Fig. 4 Left panel: Time-height crosssection of (a) attenuated backscatter coefficient, (b) Doppler spectral width, (c) horizontal wind speed, (d) horizontal wind direction, and (e) vertical wind speed over Hefei observed by CDWL between 16–17 March 2020. Right panel: simultaneous ground observation results, including (f) surface hourly particulate matter concentration from local monitor network, (g) number concentration and (h) size distribution of surface total aerosol particles from WIBS, (i) local temperature and humidity, and (j) local visibility and rain rate. The wind direction near the surface changes at about 01:30 on 17 March and is marked with a triangle symbol.**

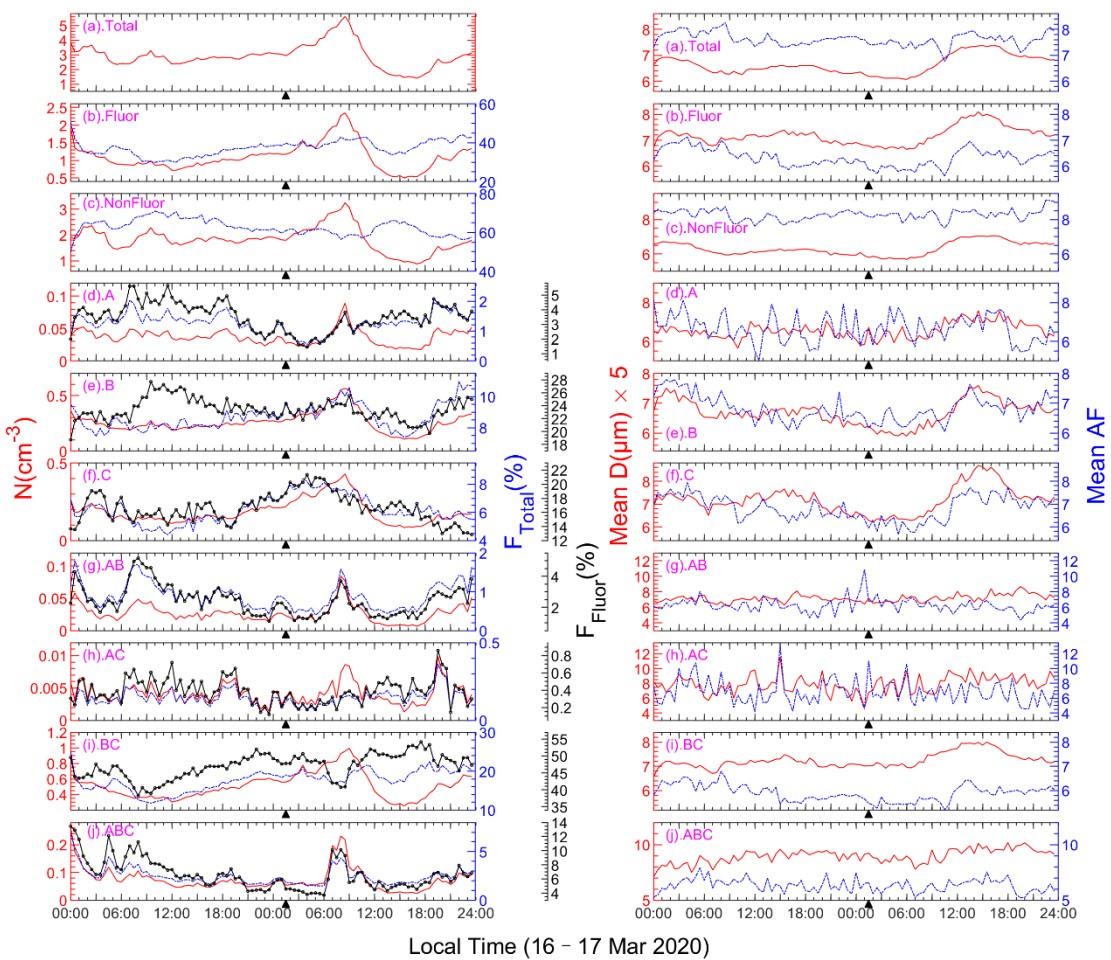

Fig. 5 Left panel: number concentrations (solid red line), number fractions to total particles (blue chain line), and number fractions to fluorescent particles (solid black line with point marker) of investigated particle types. Right panel: Count mean particle diameter (×5, solid red line) and count mean asphericity factor (blue chain line) for investigated particle types measured by WIBS on 16–17 March 2020. The direction of the wind near the surface changes at about 01:30 on 17 March and is marked with a triangle symbol.

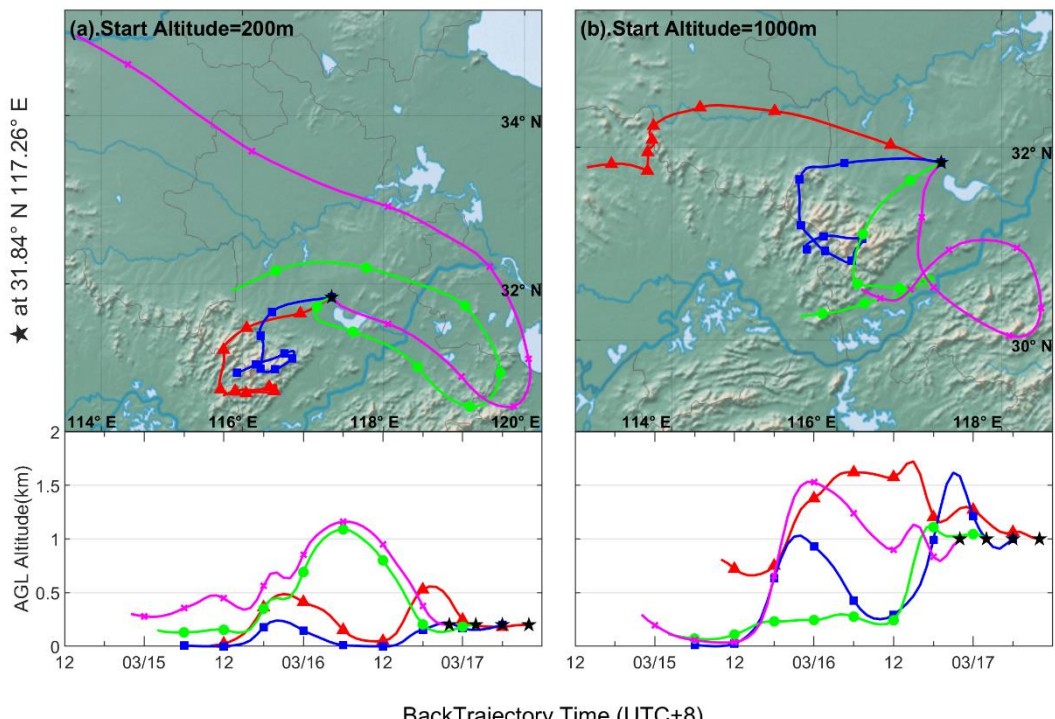


**Fig. 6 The 48 h HYSPLIT backward trajectory result calculated at 22:00 on 16 March (magenta line with cross marker), 2:00 (green line with diamond marker), 6:00 (blue line with square marker), and 10:00 (red line with triangle marker) on 17 March (UTC+8). The terminal points are marked every 6 hours. The starting location is set as 200 m (left panel) and 1000 m (right panel) over Hefei. Made with Natural Earth. Free vector**

**and raster map data @ naturalearthdata.com.**

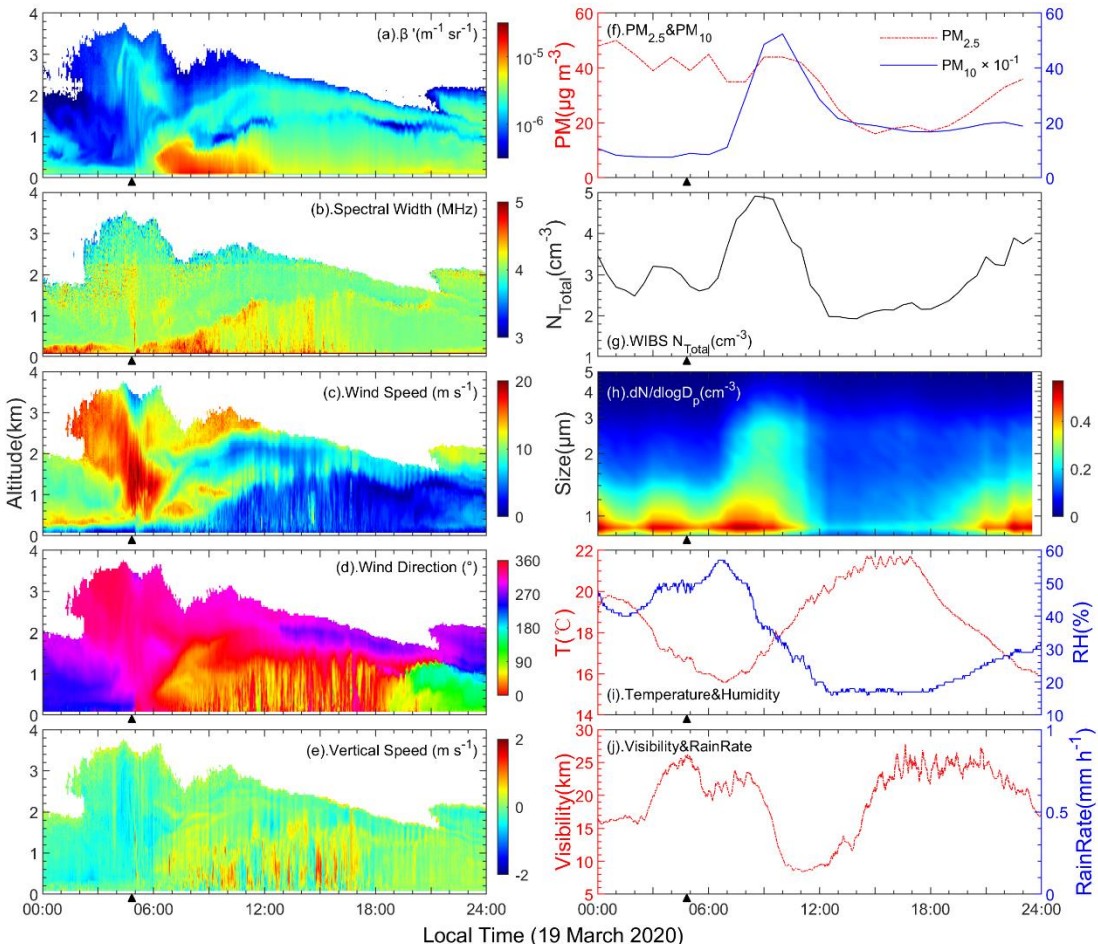

**Fig. 7 Left panel: Time-height crosssection of (a) attenuated backscatter coefficient, (b) Doppler spectral width, (c) horizontal wind speed, (d) horizontal wind direction, and (e) vertical wind speed over Hefei observed by CDWL on 19 March 2020. Right panel: simultaneous ground observation results, including (f) surface hourly particulate matter concentration from local monitor network ($PM_{10} \times 10^{-1}$ for readability), (g) number concentration and (h) size distribution of surface total aerosol particles from WIBS, (i) local temperature and humidity, and (j) local visibility and rain rate. The wind direction near the surface changes at about 05:00 on 19 March and is marked with a triangle symbol.**

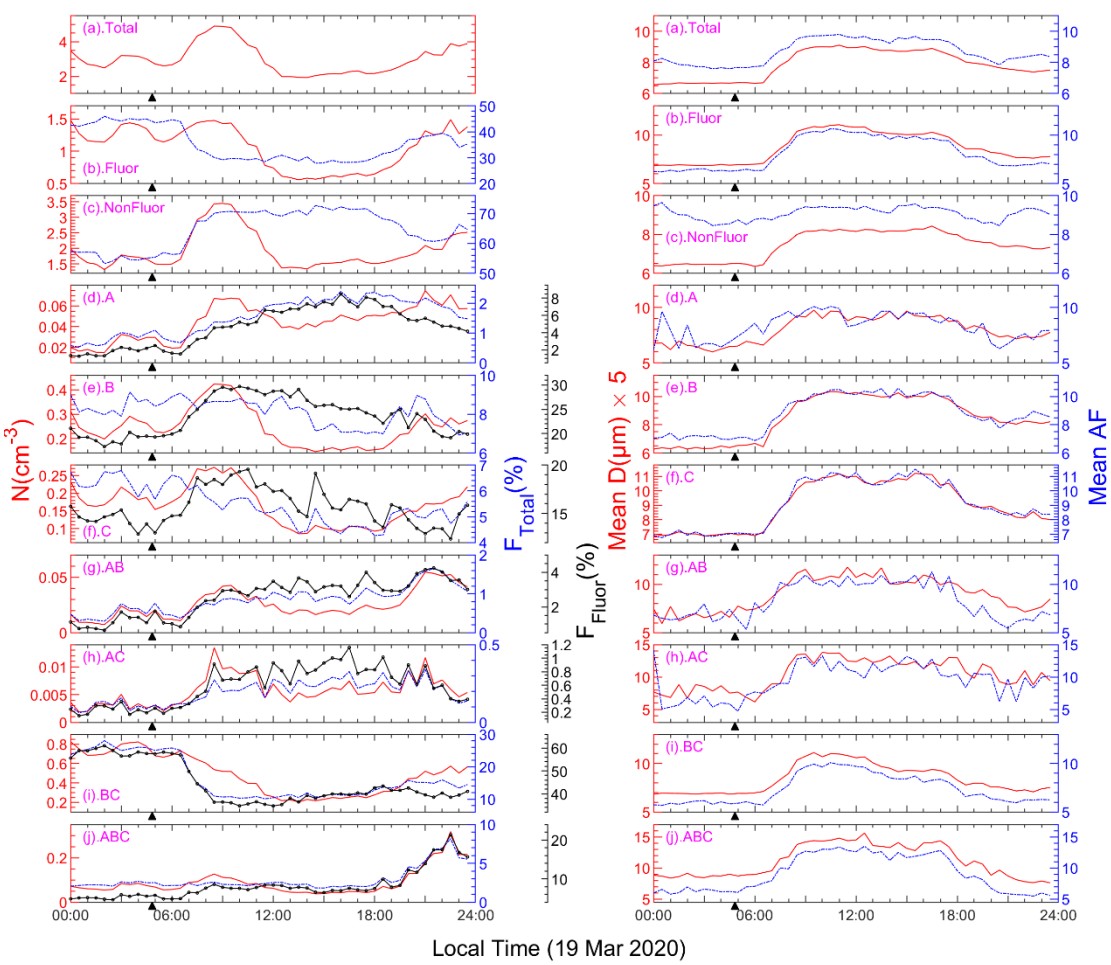

Fig. 8 Left panel: number concentrations (solid red line), number fractions to total particles (blue chain line), and number fractions to fluorescent particles (solid black line with point marker) of investigated particle types. Right panel: Count mean particle diameter (×5, solid red line) and count mean asphericity factor (blue chain line) of investigated particle types measured by WIBS on 19 March 2020. The wind direction near the surface changes at about 05:00 on 19 March and is marked with a triangle symbol.

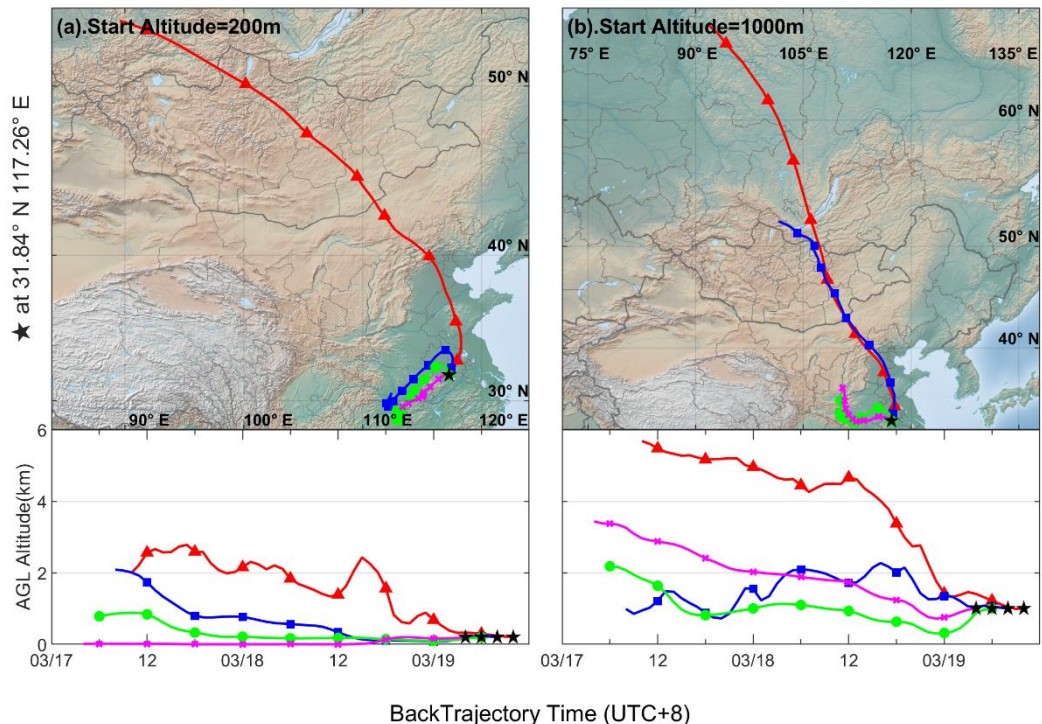


**Fig. 9 The 48 h HYSPLIT backward trajectory result calculated at 4:00 (magenta line with cross marker), 6:00 (green line with diamond marker), 8:00 (blue line with square marker), and 10:00 (red line with triangle marker) on 17 March (UTC+8). Terminal points are marked every 6 hours. The starting location is set as 200 m (left panel) and 1000 m (right panel) over Hefei. Made with Natural Earth. Free vector and raster map data @ naturalearthdata.com..**
