# Peer review of "Observation of bioaerosol transport using wideband integrated bioaerosol sensor and coherent Doppler lidar"

_Atmospheric Measurement Techniques, 2021_

## Author Comment (AC1)

We would like to thank the reviewers for their valuable comments and suggestions. We have considered all comments carefully which helped us significantly to improve our manuscript. Following the reviewers' comments and suggestions, we revised the manuscript. Our responses to the reviewers' comments are listed below in blue fonts and the changes in the manuscript are listed in *blue italic fonts*.

**General comment**

The paper uses lidar and ground level observations with a bioaerosol monitoring system to investigate the transport of bioaerosol in a measurement site in China. The analysis is performed on three case studies. The topic is interesting and suitable for the Journal, however, there are some aspects not very clear including what is the real novelty on the paper that need to be addressed in a revision step.

Thanks for your careful and thoughtful comments. We revised the manuscript according to your suggestions.

**Specific comments**

1. It is not very clear what is the novelty in the approach or in the results. This aspect should be discussed and the choice to investigate a small number of cases (three events) should be justified.

Thank you for your valuable comments. In this paper, we combine coherent Doppler wind lidar and WIBS to simultaneously detect the atmospheric phenomena and local aerosol characterization in high time resolution, to search their relationship. In the results, we observed that aerosol transport not only increased local aerosol concentration but also their composition. Besides, we observed the impact of some atmospheric phenomena, like virga, on local fluorescent aerosols in a short time, which cannot be observed by traditional off-line detection techniques. The methodology and observation results can inspire a comprehensive understanding of bioaerosol transport and its role in the atmospheric processes and aerosol-cloud -precipitation interaction. We revised the following text to emphasize the novelty of our research.

As for the small number of cases in this paper. Unfortunately, we cannot provide more events in this paper. This campaign was only performed in March 2020, during the period the aerosol transport event are rarely captured by lidar, and some suspected events are obscured in lidar observation due to the simultaneous bad weather condition (precipitation and cloudy days often occur in Hefei in Spring). Furthermore, aerosol transport and precipitation can both influence the local bioaerosol. When aerosol transport and precipitation occur at the same time, the variation of WIBS data cannot be attributed to whether aerosol transport or precipitation. These three transport events are presented in the paper because of their good observation quality in good weather conditions during the time. And the characteristics, such as the difference of wind direction in different altitudes and the high PM concentration near the ground also help the confirmation of aerosol transport events. Observations using lidar and WIBS in other times and seasons will be performed in the future.

**Changes: lines 23-30.**

*"The results prove the influence of external aerosol transport on local high particulate matter (PM) pollution and fluorescent aerosol particle composition. The combination of WIBS and CDWL expands the aerosol monitoring parameters and provides a potential method for real-time monitoring of fluorescent biological aerosol transport events. In addition, it also helps to understand the relationships between atmospheric phenomena at high altitudes like virga and the variation of surface bioaerosol. It contributes to the further understanding of long-range bioaerosol transport, the roles of bioaerosols in atmospheric processes and in aerosol-cloud-precipitation interactions."*

**Changes: line 105-118.**

*"This paper provides a new perspective for the study of bioaerosol transport. CDWL enables continuous monitoring of multiple atmospheric parameters in real time, such as aerosol extinction coefficient, wind vector, turbulence activity, precipitation, etc. Based on LIF technologies, WIBS provides detailed single-particle information containing up to 5 parameters. It provides a higher time resolution monitoring of aerosols compared with traditional offline measurement methods based on sampling analysis, and compared with online aerosol measurement instruments such as particle sizer, it expands the dimension of aerosol measurement and thus enables categorized monitoring of aerosol. The combination of these two instruments helps to understand the potential impact of external bioaerosols on local bioaerosol composition during aerosol transport. In addition, lidar is capable of detecting atmospheric phenomena in high altitudes like the virga, which cannot be measured by ground-based in-situ measurement instruments, and thus enables discovering their relationships with the variation of bioaerosols at ground level during this event. The phenomenon suggests that the combination of these two instruments also contributes to a further understanding of the role of bioaerosols in atmospheric processes and in aerosol-cloud-precipitation interactions."*

2. Lines 48-51. These aspects should be backed up with a references. In addition, it could strongly depend on the type of bioaerosol with bacteria and fungi behaving differently from small viruses in terms of probability of attachment on pre-existing particles. I believe that this should also be mentioned.

Thank you for your suggestion, additional references have been added and the different behavior in survival mechanisms among different types of bioaerosols has been mentioned.

**Changes: lines 50-56.**

*" To survive in dry and intense solar radiation environments at high altitudes during long-range transport, bioaerosols have developed some survival mechanisms including pigment deposition (Tong and Lighthart, 1997), sporulation (Griffin, 2007), and attaching themselves to other particles like dust (Griffin et al., 2001). These survival mechanisms depend on the type of bioaerosol, for instance, due to their relatively small size compared with bacteria and fungi, viruses are more likely to be attached to other large pre-existing particles under the influence of Brownian motion."*

3. Another aspect that should be mentioned is that the WIBS is not able to measure small particles (< 500 nm) so that the methodology used is focused on relatively coarse bioaerosol with limited potentiality for viruses for example.

Thank you for your mention, additional descriptions about the limitation of WIBS have been added.

**Changes:, lines 78-82.**

*" However, WIBS still has some limitations in detecting bioaerosols. For example, WIBS cannot detect particles whose size is smaller than 0.5 μm, so WIBS has limited potential in the detection of small size bio-particles like viruses and focuses on relatively coarse bioaerosols. Besides, non-biological fluorescent components on aerosol particles, such as polycyclic aromatic hydrocarbons (PAHs), humic acids, and fulvic acids may act as interferent during WIBS measurements."*

4. Lines 294-301. Percentages of what of total particles or of fluorescent particles?

Thank you for your comments. Here we describe the fraction of a specific kind of fluorescent particles in whole fluorescent particles. For instance, $F_{Fluor}(BC) = \frac{N_{BC}}{N_{Fluor}} = \frac{N_{BC}}{N_A+N_B+N_C+N_{AB}+N_{AC}+N_{BC}+N_{ABC}}$. In WIBS data, based on the different fluorescent intensities in three fluorescent channels, the whole fluorescent particles (Marked as Fluor) are further categorized into seven types of fluorescent particles(Perring et al., 2015). The related plot is portraited in the black line in the left panel in Fig 2,5,8. An additional description is added and an appendix is added to describe the abbreviation and parameters used in this text.

**Changes:**

**Line 371-381.**

*"The statistical results of WIBS data between 16–17 March are shown in Fig. 5. The number concentration of each type of aerosol increases during the transport event (Fig. 4). Their maximum concentrations are all observed at about 8:30, which is consistent with the time of maximum $PM_{10}$ concentration (Fig. 4(f)). While number concentrations increase, different types of fluorescent aerosol particles show different trends in their number fraction to whole fluorescent aerosol particles. For example, $F_{Fluor}(BC)$ shows a significant drop from 48.4 % at 6:00 to 40.9 % at 8:30 (Fig. 5 (i)) but $F_{Fluor}(A)$, $F_{Fluor}(AB)$ and $F_{Fluor}(ABC)$ increase from 1.8 %, 1.6 % and 3.7 % at 6:00 to 3.8 %, 3.7 % and 9.3 % at 8:30 on 17 March respectively (Fig. 5 (d)(g) and (j)). These changes that occur in a short amount of time reveal that the transport of aerosols not only leads to high $PM_{10}$ and $PM_{2.5}$ concentrations but also leads to the increase of some types of fluorescent particles in their fraction to whole fluorescent particles."*

**Line 549-550**

*"*

*Appendix A*

*Table A1. Description of abbreviations and parameters in the text*

| Abbreviation | Description |
|---|---|
| Total | Total aerosol particles detected by WIBS |
| Fluor | Fluorescent aerosol particles detected in any one of the three channels |
| Type FL-1 | Fluorescent aerosol particles detected in channel FL1 |
| Type FL-2 | Fluorescent aerosol particles detected in channel FL2 |
| Type FL-3 | Fluorescent aerosol particles detected in channel FL3 |
| Type A | Fluorescent aerosol particles detected in channel FL1 only |
| Type B | Fluorescent aerosol particles detected in channel FL2 only |
| Type C | Fluorescent aerosol particles detected in channel FL3 only |
| Type AB | Fluorescent aerosol particles detected in channels FL1 and FL2 |
| Type AC | Fluorescent aerosol particles detected in channels FL1 and FL3 |
| Type BC | Fluorescent aerosol particles detected in channels FL2 and FL3 |
| Type ABC | Fluorescent aerosol particles detected in channels FL1, FL2, and FL3 |
| $N_{xx}$ | The number concentration of a certain type of aerosol particles |
| $F_{Total}(xx)$ | Number fraction to total aerosol particles of a certain type of aerosol particles, i.e. $N_{xx}/N_{Total}$ |
| $F_{Fluor}(xx)$ | Number fraction to fluorescent aerosol particles of a certain type of aerosol particles, i.e. $N_{xx}/N_{Fluor}$ |
| Mean $D_{xx}$ | Count mean particle diameter of a certain type of aerosol particles detected by WIBS |
| Mean $AF_{xx}$ | Count mean asphericity factor of a certain type of aerosol particles detected by WIBS |

*"*

5. Lines 299-301. Actually the peaks in PM2.5 and PM10 seems to be more related to the increase of the fraction of non fluorescent particles rather than of fluorescent at least according to Figures 2a, 2b, and 2c. This sentence should be explained better or corrected.

Thanks for your comments. Actually, the text in Line 299-301 describes Fig. 5. As shown in Fig. 5, when $PM_{2.5}$ and $PM_{10}$ reach their peak, there is a slight increase in the fraction of fluorescent particles to total particles (Fig. 5(b), left panel, blue line). The main reason should be the increasing concentration of Type A, AB, and ABC particles (Fig. 5(d) (g) (j)), according to the variations in the concentration of different types of aerosol.

In Fig.2, as you described, the fraction of non-fluorescent particles does increase when $PM_{2.5}$ and $PM_{10}$ reach their peak on March 13, and the non-fluorescent are more related to $PM_{2.5}$ and $PM_{10}$. The reasons for these phenomena are discussed in sec.3.1.3 of the revised version. However, in March 13 event, we focus on the different behavior of Type AB and ABC aerosols from other types of aerosols before PM

concentrations reach their peak. The increase of non-fluorescent aerosol when PM concentration reaches their peaks do not disturb the conclusion. The explanation has been modified to avoid confusion.

**Changes:**

**line 277-282:**

*"However, Type AB and Type ABC particles have different time variation trends from the above-mentioned fluorescent particles: $N_{AB}$ and $N_{ABC}$ both start to increase at about 3:30 when the wind near the surface changes (Fig. 2(b)) and they reach their peak at about 7:00–8:00, causing $F_{Fluor}(AB)$ and $F_{Fluor}(ABC)$ to increase from 1.6 % and 7.2 % to their maximum of 6.5 % and 16.9 % respectively, and consequently leading $F_{Total}(Fluor)$ to increase to its daily maximum of 34.7 % at 6:30."*

**line 315-323:**

*"At 12:00, when the $PM_{2.5}$ concentration in Hefei reaches its maximum, both the wind near the surface and at high altitude show a transport path passing several coastal cities in Yangtze River Delta Urban Agglomerations such as Hangzhou, Ningbo, and Shanghai, which can be a source of wet air mass and pollutants. The transported pollutants, the increasing emission of local anthropogenic aerosol after sunrise, the accumulation of aerosols caused by the low-altitude cloud layer, and the hygroscopic growth of small size aerosol particles under high humidity altogether contribute to the rapid increase of PM concentration. These increased aerosol particles have a less fraction of fluorescent biological aerosols, which explains the rapid decrease of fluorescent particles in their fraction to total particles observed by WIBS after 7:30."*

**line 379-381:** *"These changes that occur in a short amount of time reveal that the transport of aerosols not only leads to high $PM_{10}$ and $PM_{2.5}$ concentrations but also leads to the increase of some types of fluorescent particles in their fraction to whole fluorescent particles."*

6. Looking at Figure 8, it seems that most of the increase seen in PM2.5 is actually due to non fluorescent particles. How it is explained this aspect or how this is in agreement with a transport of bioaerosol?

Thank you for your valuable comments. Indeed, there is a decrease in the fraction of fluorescent particles when PM concentration increases. And the main reason is the abnormal concentration decrease of Type BC particles, which can be proved by the similar magnitude of decrease in $F_{Total}(BC)$ (-15.2 % during event) and $F_{Total}(Fluor)$ (-15.5 % during event). However, when considering the fraction to total particles, to exclude the effect of decreased $N_{BC}$, $F_{Total}(A)$ and $F_{Total}(AB)$ actually increased during the event. Therefore, the external transported aerosols have lower fractions of Type BC particles and fluorescent particles but higher fractions of Type A and Type AB particles.

The typical related biological source of Type A, AB, BC particles has been discussed in the revised manuscript. Laboratory research (Hernandez et al., 2016) shows that Type FL-1 (A, AB, ABC) is often connected with bacteria and fungi. Although some pollen is categorized into Type BC particles, it is hard to think of such a high concentration and fraction of pollen ($F_{Fluor}$ ($BC$)> 50 % before event) in urban area. A better explanation of Type BC particles is the highly-oxygenated humic-like substances (HULIS), which belongs to non-biological interferents and has a similar fluorescent fingerprint of Type BC particles (Chen et al., 2016, 2020; Wen et al., 2021; Yue et al., 2019) and show moderate correlation with Type BC particles in field campaign (Yue et al., 2019). The high concentration of highly-oxygenated HULIS in the urban area and its decrease during dust transport are also observed in research (Wen et al., 2021).

**Changes: line 435-455, sec. 3.3.2**

*"Figure 8 reveals that as the external aerosol reaches the ground (Fig. 7(a)), $\mathrm{Mean\ D}$ and $\mathrm{Mean\ AF}$ of all types of fluorescent aerosol particles increase sharply at different degrees from 6:00 on 19 March, which is higher than that in the two events described before and also the highest record during the observation period. The impact from the external aerosol indicates that the external aerosol layer is dominated by non-spherical aerosol particles in coarse mode. Although all other types of fluorescent particles exhibit an increase in their number concentration during this event, Type BC shows a different trend. As portrayed in Fig. 8(i), $N_{BC}$ abnormally decreases from 0.68 cm⁻³ at 6:00 to 0.52 cm⁻³ at 9:00, causing $F_{Fluor}$ ($BC$) to decrease from 57.6 % at 6:00 to 36.4 % at 9:00 and $F_{Total}$ ($BC$) to decrease from 25.8 % at 6:00 to 10.6 % at 9:00. Considering its high concentration before 6:00, the sharp decline of $N_{BC}$ also lead to the rapid decrease of $F_{Total}$ ($Fluor$), which decrease from 44.7 % at 6:00 to 29.2 % at 9:00 (Fig. 8(b)). To compensate for the decreased $N_{BC}$, all other types of fluorescent particles show increases in their $F_{Fluor}$ in different degrees, which is predictable. However, when considering the $F_{Total}$ of other types of fluorescent particles to exclude the influence of decreased $N_{BC}$, they do not exhibit sharp decreases in $F_{Total}$ like Type BC particles does, only $F_{Total}$ ($C$) shows slightly decrease from 6.6 % to 5.3 % (Fig. 8(f)). On the contrary $F_{Total}$ ($A$) and $F_{Total}$ ($AB$) increase from 0.7 % and 0.5 % at 6:00 to 1.4 % and 0.9 % (Fig. 8 (d) and (g)). From the almost the similar decline range of $F_{Total}$ ($BC$) and $F_{Total}$ ($Fluor$) and no similar sharp decrease of $F_{Total}$ observed in other types of fluorescent particles, it can be concluded that the major reason for the significant drop of $F_{Total}$ ($Fluor$) are the abnormal decrease of $N_{BC}$ during the event. In addition, the increase of $F_{Total}$ ($A$) and $F_{Total}$ ($AB$) during transport indicates that the external transported aerosol has a higher fraction of Type A and Type AB particles and a much lower fraction of Type BC particles than local aerosols."*

**Changes: line 501-516**

*"In addition, the abnormal decrease in the concentration of Type BC particles should also be noted during this event. Laboratory test (Hernandez et al., 2016) shows that Type BC particles are not a typical fluorescent type for bacteria and fungal spores. Bacteria and fungal spores are mainly connected with Type FL-1 particles. On contrary, EEM (excitation-emission matrix) analysis result from sampled aerosol (Chen et al., 2016, 2020; Wen et al., 2021; Yue et al., 2019) shows that highly-oxygenated HULIS (humic-like substances) has a similar fluorescent fingerprint to Type BC particles and can thus be non-biological fluorescent interferent during WIBS measurement by categorized into Type BC particles. This point is supported in an Mt. Tai field campaign (Yue et al., 2019) by the moderate correlation between*

*Type BC particles and highly-oxygenated HULIS. Atmospheric HULIS has many sources including primary emission mechanisms like biomass burning, and multiple secondary formation processes (Wu et al., 2021). Compared with rural areas, there are abundant air pollutants of strong oxidation capacity in urban areas like NOx and O3, which may contribute to the formation of highly-oxygenated organic aerosol like HULIS (Tong et al., 2019). This can explain the high concentration and fraction of local Type BC particles and the abnormal decrease of its concentration and fraction during external aerosol transport. This explanation is adopted by previous research (Wen et al., 2021), where they also find the decrease of highly-oxygenated HULIS concentration in the urban area during the dust period."*

7. Lines 342-345. It should be explained better why it is believed that in this case there was bioaerosol attached to pre-existing dust and not in the other cases. A similar question arise for the lines 360-367. The difference with the previous mentioned work what that ratio is compared for aged and local aerosol could be due to differences in the local sources rather than on bioaerosol attached to dust. Please explain and comment better these points.

Thank you for your valuable comments. Judging from the transport path, the high $PM_{10}$ concentration during the event, and the characteristics of external aerosols which are unseen in the previous two events detected by WIBS, the external aerosols are most likely to be mineral dust. Compared with local fluorescent aerosols, external fluorescent aerosols have higher sizes but lower fluorescent quantum efficiency. So it is not appropriate to assume these transported fluorescent aerosols are individual bioaerosol particles. A better explanation is that these transported external large-size particles result from the bacteria attached to dust. Detailed discussions are contained in the changes of the text.

Besides, we abolished the previous explanation that the different ratio of $(N_A + N_{AB})/N_{BC}$ between local and external aerosols are attributed to the survival mechanism of bioaerosol by attaching to other particles. As discussed in Comment 6, the highly-oxygenated HULIS is related to Type BC particles in WIBS detection. There is new evidence showing that highly-oxygenated HULIS have many sources including primary emission and secondary formation. The high abundant Type BC particles in Hefei should be attributed to the high abundant air pollution of strong oxidation capacity in urban rather than a transformation from bioaerosol. And the abnormal decrease of Type BC particles during the event should be attributed to the different compositions of external and local aerosols. Detailed discussion and additional references are covered in the changes in Comment 6.

**Changes: line 465-500**

*"The increased mean diameter and asphericity factor of all types of aerosol particles, the transport path, and the high $PM_{10}$ concentration during the event altogether indicate the transported external aerosols are dominant by large-size non-spherical particles, which are most likely to be mineral dust. Although $F_{Total}(Fluor)$ decreases during the transport event, the increasing $F_{Total}(A)$ and $F_{Total}(AB)$ during the transport event still indicates the potential bioaerosol transport, because bacteria and fungal spores are related with Type FL-1 (usually Type A, AB, ABC particles) particles (Hernandez et al., 2016; Savage et al., 2017). According to their mean diameter, the sizes of these transported Type A and Type AB particles during this event are mainly distributed in the range of larger than 1.8 μm for Type A*

*particles and larger than 2.0 μm for Type AB particles, which means the potential transported bioaerosol particles are the largest among the three transport events. However, it is not appropriate to assume that these increased Type A and Type AB are individual bioaerosol particles. Previous works (Savage et al., 2017; Yue et al., 2019) show that larger fluorescent biological aerosol particles are more likely to exhibit a higher fluorescent intensity and wider fluorescent spectrum range, which means exhibit fluorescent in multiple fluorescent channels in WIBS measurement. For example, larger size fungal spores and pollen have higher probabilities to be categorized as Type ABC particles than smaller bacteria. During this event, the transported external aerosol particles have larger sizes than local aerosol particles, however, the expected particles with the highest fluorescent intensities, Type ABC particles do not show an obvious increase in their fraction to total particles. These phenomena indicate that the larger size transported external fluorescent particles do not have an apparently higher fraction of Type ABC particles, and thus have lower fluorescent efficiency than local fluorescent particles. In addition, considering the transported external Type A and Type AB particles have the particle size and asphericity factor characteristics like mineral dust, it is a better explanation that these transported external large-size Type A and Type AB particles result from the bacteria that attached to dust. Researches reveal that microbial activity is significant in the aerosols from desert regions, even impacting the composition of aerosols in downwind regions (Ho et al., 2005; Hua et al., 2007; Maki et al., 2014, 2015; Tang et al., 2018). During long-range transport, larger mineral particles attached by bacteria can serve as a shelter and favor the survival of bacteria. Dust-attached bacteria have been found in SEM (scanning electron microscope) images from air samples of previous research (Tang et al., 2018). A field campaign (Maki et al., 2019) in the Gobi Desert, which is the source of this event, reveals that after dust events, bacteria from Bacteroidetes, which are known capable of attaching to coarse particles, increase their relative abundance in air samples. The above results support our explanation of WIBS data. The pathogenic bioaerosol during dust transport events is believed to be linked to allergen burden and asthma (Ichinose et al., 2005; Liu et al., 2014), and even multiple diseases such as Kawasaki disease in humans (Rodó et al., 2011) and rust diseases in plants (Fröhlich-Nowoisky et al., 2016). Dust transport events generally occur during spring in Hefei. The WIBS data during this event indicates that the long-range dust transport during spring has potential risks to human health in the Hefei area."*

8. Line 345. Ok for diffusion, including vertical diffusion due to the increasing boundary-layer depth, however, I do not see any evidence of dry deposition in this dataset.

Thank you for your suggestion. The high $PM_{10}$ concentration during this event indicates that there are a large number of large size particles. Usually, large particles tend to settle faster by gravity (Valsaraj and Kommalapati, 2009). But this process lacks direct observation evidence during this event. For a more rigorous explanation, the text has been modified and the description about dry deposition has been removed.

**Changes: line 427-433**

*"WIBS data (Fig. 7 (g) (h)) show that particles in coarse mode are most abundant during this period. Temperature is rising and humidity decreasing (Fig 7. (i)) while $PM_{10}$ is increasing, which inhibits hygroscopic growth and accumulation of aerosol particles. So, it can be inferred that the sharp increase*

*of PM$_{10}$ concentration cannot be attributed to the local aerosol accumulation or hygroscopic growth, but the transport of external aerosols. After sunrise, with an increase in solar radiation, the PBL height rises, and aerosol diffusion increases. The increased aerosol diffusion contributes to the decrease of PM concentration after 10:00."*

9.  Line 402. See the previous comment on attached dust, if this is not the only explanation please be more cautious in interpretation.

Thank you for your suggestion. Based on the reply in comment 7. We decide to remain the text here.

10. Line 28. Probably it is better to say long-range bioaerosol transport.

Thanks for your suggestion. We correct it.

11. Line 330. Use an apex for m s$^{-1}$.

Thanks for your suggestion. We correct it.

**References**

[revised manuscript text omitted]

---

## Author Comment (AC2)

We would like to thank the reviewers for their valuable comments and suggestions. We have considered all comments carefully which helped us significantly to improve our manuscript. Following the reviewers' comments and suggestions, we revised the manuscript according to your suggestions. Our responses to the reviewers' comments are listed below in blue fonts and the changes in the manuscript are listed in *blue italic fonts*.

This paper presents a study on the transport of bioaerosols, examining three case studies occurred in March 2020 at Hefei (China). The aerosols are also discriminated into fluorescent and non-fluorescent particles by using a wideband integrated bioaerosols sensor (WIBS) measurements. Also, the biological fluorescent particles are typed into several categories by applying previous methodologies. In combination with a Doppler wind lidar measurements and HYSPLIT back-trajectory modelling their transport is investigated. The relevance of this work can rely on the use of WIBS in-situ instrument in synergy with the Doppler wind lidar (remote sensing) for bioaerosol transport studies, but the novelty of this work is not clearly appreciated.

Therefore, this work could be published, but the following comments should be addressed before it is accepted for publication in AMT.

Thanks a lot for your comments and encouragement. We revised the manuscript according to your comments and suggestion.

**General comments:**

1. Regarding bioaerosols, although it is mentioned (page 3, lines 73-75), there is no discussion on the potential errors in the WIBS measurements due to the non-biological component of the registered fluorescent particles, whose concentration could be lower than that reported in the paper. Please, include such a discussion.

   Thank you for your comment. Additional discussions about the potential errors introduced by fluorescent interferents are added in the revised manuscript.

   **Changes: line 186-196**

   *"As mentioned before, non-biological fluorescent components on aerosol particles can be fluorescent interferent during WIBS observation. So, the concentrations of fluorescent aerosol particles can be higher than the actual concentration of local fluorescent biological aerosol particles. Proper fluorescent threshold configuration can eliminate these non-biological interferents as much as possible and remain biological particles categorized into fluorescent as many as possible. Although laboratory test (Savage et al., 2017) shows that applying a higher threshold like 6σ or 9σ threshold can effectively exclude interferent like wood smoke and brown carbon from being categorized into fluorescent, filed campaign (Yue et al., 2019) suggest that a proportion of bioaerosols can be misclassified into non-fluorescent particles when elevating*

*threshold. The 3σ threshold strategy adopted in this paper is to keep consistent with previous works (Crawford et al., 2015; Yu et al., 2016; Yue et al., 2016)."*

2. About the type of the bioaerosols more predominantly found in each of the three events examined, it is necessary to provide a more complete explanation in the discussion and justification in the conclusions. Moreover, it must be included a correspondence (maybe in a Table) between each of the WIBS fluorescent particle categories (A, B, C, …, ABC), together with their main fluorescence characteristic/parameters), and their most likely associated type of bioaerosol (fungi, bacteria, pollen, …), in addition to the corresponding, already provided, references. The reading of the paper will indeed be improved.

Thank you for your valuable advice. A table of typical aerosol characterization in WIBS is added as an appendix for reference. And the discussions of possible transported bioaerosol are all improved for a more complete explanation and a more justified conclusion. There are several changes related to the discussion. Parts of them are listed in the following text. Complete changes can be seen in the revised manuscript.

**Changes: line 324-344,**

[revised manuscript text omitted]

*"*

3. Three case studies, occurring all for a short-time period (11-20 March 2020), seem to be a short sampling for the evaluation of bioaerosol transport over Hefei. Please, provide other events (in other seasons, for instance). In particular, the last dust-bioaerosol case should include an extra discussion regarding the relevance of the potential pathogenic biological targets being transported on dust intrusions, for instance, as Hefei (China) is a frequent Asian dust-influenced zone.

Thank you for your valuable suggestions. But unfortunately, we cannot provide more events in this paper. This campaign was only performed in March 2020, during the period the aerosol transport event are rarely captured by lidar, and some suspected events are obscured in lidar observation due to the simultaneous bad weather condition (precipitation and cloudy days often occur in Hefei in Spring). Furthermore, aerosol transport and precipitation can both influence the local bioaerosol. When aerosol transport and precipitation occur at the same time, the variation of WIBS data cannot be attributed to whether aerosol transport or precipitation. These three transport events are presented in the paper because they can be confirmed by the good observation quality in good weather conditions during the time, the difference of wind direction in different altitudes, and the high PM concentration near the ground. Observations using lidar and WIBS in other times and seasons will be performed in the future.

Your suggestion for extra discussion regarding the relevance of the potential pathogenic biological targets being transported on dust intrusions is considered in the revised version. Additional discussion about the dust transport and the resulting health risk is added.

**Changes: line 495-500**

*"The pathogenic bioaerosol during dust transport events is believed to be linked to allergen burden and asthma (Ichinose et al., 2005; Liu et al., 2014), and even multiple diseases such as Kawasaki disease in humans (Rodó et al., 2011) and rust diseases in plants (Fröhlich-Nowoisky et al., 2016). Dust transport events generally occur during spring in Hefei. The WIBS data during this event indicates that the long-range dust transport during spring has potential risks to human health in the Hefei area."*

4. Unfortunately, the WIBS instrument only register particles with sizes > 0.5 microns (> 0.8 microns as discussed in the paper). This fact should be highlighted in the paper, as WIBS measurements only can register mostly coarse bioaerosols, missing the contribution of smaller-sized bio-particles.

Thank you for your suggestion. Additional discussion about the limit of WIBS is added in the revised version.

**Changes: Page 3, line 78-82**

*"However, WIBS still has some limitations in detecting bioaerosols. For example, WIBS cannot detect particles whose size is smaller than 0.5 μm, so WIBS has limited potential in the detection of small size bio-particles like viruses and focuses on relatively coarse bioaerosols. Besides, non-biological fluorescent components on aerosol particles, such as polycyclic aromatic hydrocarbons (PAHs), humic acids, and fulvic acids may act as interferent during WIBS measurements."*

**Other comments:**

1. Page 2, line 29: Not only 'fine solid particles', but also coarse solid particles. Please, remove 'fine'.

Corrected as suggested.

**Changes: line 31**

*"Aerosols are suspensions of solid particles or liquid droplets in the atmosphere "*

2. Page 4, Section 2.1: Does the wind direction at an angle of 0º corresponds to winds from the North? Does it increase clockwise or opposite? Please, indicate it.

Thank you for your question. Yes, the wind direction at 0° corresponds to winds from the north and increase clockwise. Additional description is added.

**Changes: line 134-135**

*"In this paper, the wind direction of 0° corresponds to horizontal wind from the north, and the angle increases clockwise."*

3. Pages 4-7 (Section 2): Please, provide the distance between the diverse instrumentation used. This can give a perspective for potential discrepancies due to the different atmospheric samplings of both the in-situ and remote sensing measurements are carried out.

Thank you for your suggestion. All instruments except the PM monitoring instruments are located on the campus of the University of Science and Technology of China (USTC, 31.84 °N,117.26 °E). Lidar system is located about 100 m west of the building of the School of Earth and Space Science. WIBS and other meteorological instruments are located in the building of the School of Earth and Space Science building. WIBS are located in a room on the top floor of the building and other instruments are on the rooftop of the building. The distance among these in-situ measurement instruments is less than 30 m. The PM monitoring stations are spread all over Hefei, the nearest

station from our lidar is about 2.7 km northwest of the campus. The information about the locations is added in the revised version.

**Changes: line 128-131**

*"The lidar observation is performed on the campus of the University of Science and Technology of China (USTC, 31.84 °N,117.26 °E) located in the urban area of Hefei, Anhui Province. The lidar system is placed on the grassland about 100 m west away from the building of the School of Earth and Space Science (SESS)."*

**Changes: line 152-158**

*"During the observation in Hefei, hourly PM concentration data published by the Department of Ecology and Environment of Anhui Province (https://sthjt.ah.gov.cn/) are used to be compared with the lidar observation and surface aerosol concentration measured by WIBS. These data are comprehensive values from the measurement results of multiple stations in Hefei, whose locations are shown in https://aqicn.org/city/hefei/. The nearest station to our lidar is located on Changjiang Middle Road (31.852 °N, 117.25 °E), about 2.7 km northwest of the USTC campus."*

**Changes: line 161-162**

*"These instruments are located on the rooftop of the SESS building."*

**Changes: line 217-218**

*"During observation, a WIBS instrument (NEO model) is located in a room on the top floor of the SESS building. within 30 m away from those meteorological instruments mentioned before."*

4. Page 8, lines 200-203: Please, give the position where the measurements are performed with respect to Hefei.

Thank you for your suggestion. The measurements are performed in the USTC campus (31.84 °N,117.26 °E) located in the urban area of Hefei, about 3 km southwest of the city center. The information is added in the revised version.

**Changes: line 128-129**

*"The lidar observation is performed on the campus of the University of Science and Technology of China (USTC, 31.84 °N,117.26 °E) located in the urban area of Hefei, Anhui Province."*

5. Page 8, lines 211-214: Please, justify extensively this statement.

Thank you for your suggestion. The statement here is justified for a more cautious explanation.

**Changes: line 249-261**

*"When the cloud layer stays at the low-altitude layer between 9:00 and 21:00, local weather conditions (Fig. 1(i)) show high humidity (78 %–89 %) and low temperatures (6 ℃–13 ℃), which inhibit aerosol diffusion but favor the accumulation of local aerosols and hygroscopic growth of particles. As such, the increase in particulate matter concentration from 7:00 to 9:00 is mainly attributed to external aerosol transport. The different trend between $PM_{2.5}$ and $PM_{10}$ concentrations after 10:00 can be attributed to their different removal efficiency: large particles have higher hygroscopic growth factors (Haenel et al., 1978; Hänel, 1976), their sizes increase largely through hygroscopic growth, and thus they have a higher probability to be removed by gravitational settlement than smaller particles. As for $PM_{2.5}$, they have lower removal efficiency and their concentration can continue to increase through the intensifying local anthropogenic aerosol emission after morning and the inhibited aerosol diffusion. During the period, $PM_{2.5}$ is observed to exceed the $PM_{10}$ concentration at one time, which can be attributed to the difference between instruments and methods for monitoring $PM_{2.5}$ and $PM_{10}$."*

6. Page 8, lines 220-221: '… due to the difference in observation location …', please, add the distance each other.

Thank you for your suggestion. The distance between WIBS and the nearest PM monitoring station is about 2.7 km. The distance is added in the revised version.

**Changes: line 267-270**

*"The different behavior of PM data and WIBS data may be due to the difference in observation location and monitoring method (As mentioned in Sec.2.2., the nearest PM monitoring location is about 2.7 km northwest of the USTC campus)."*

7. Page 10, lines 276-277: Any reference or explanation is needed to justify this sentence.

Thank you for your suggestion. Additional explanations and references are added in the revised version.

**Changes: line 351-356**

*"In previous works, a fall velocity greater than 1 m s$^{-1}$ can be identified as precipitation (Manninen et al., 2018; Wang et al., 2019a) and Doppler spectral will be broadened during precipitation due to additional signal peaks from raindrops (Wei et al., 2021). Although these characteristics are the same as a precipitation event, no rainfall is recorded on the ground during this event (Fig.4(j)).*

*These observations indicate a virga event, during which the hydrometeors beneath the cloud layer enhance lidar backscatter and evaporate before reaching the ground."*

8.  Page 13 lines 363-366: A more complete justification/explanation is required for this statement.

Thank you for your suggestion. The context here is totally modified. We abolished the previous explanation that the different ratio of $(N_A + N_{AB})/N_{BC}$ between local and external aerosols are attributed to the survival mechanism of bioaerosol by attaching to other particles. The highly oxygenated HULIS related to Type BC particles have sources including primary emission mechanism and secondary formation. In addition to the previous-mentioned transformation from bioaerosol by photo-oxidation, the abundant air pollution of strong oxidation capacity in urban areas like $NO_x$ and $O_3$ can contribute to the formation of highly-oxygenated organic aerosol like HULIS. However, in desert areas, these air pollutants are less abundant, which means the main generation mechanism of HULIS can be different between local aerosol and external aerosol. And this can explain the higher concentration and fraction of local Type BC aerosols than external aerosols during observation and its decrease of concentration and fraction during external aerosol transport.

**Changes: line 501-516**

"

*In addition, the abnormal decrease in the concentration of Type BC particles should also be noted during this event. Laboratory test (Hernandez et al., 2016) shows that Type BC particles are not a typical fluorescent type for bacteria and fungal spores. Bacteria and fungal spores are mainly connected with Type FL-1 particles. On contrary, EEM (excitation-emission matrix) analysis result from sampled aerosol (Chen et al., 2016, 2020; Wen et al., 2021; Yue et al., 2019) shows that highly-oxygenated HULIS (humic-like substances) has a similar fluorescent fingerprint to Type BC particles and can thus be non-biological fluorescent interferent during WIBS measurement by categorized into Type BC particles. This point is supported in an Mt. Tai field campaign (Yue et al., 2019) by the moderate correlation between Type BC particles and highly-oxygenated HULIS. Atmospheric HULIS has many sources including primary emission mechanisms like biomass burning, and multiple secondary formation processes (Wu et al., 2021). Compared with rural areas, there are abundant air pollutants of strong oxidation capacity in urban areas like NOx and O3, which may contribute to the formation of highly-oxygenated organic aerosol like HULIS (Tong et al., 2019). This can explain the high concentration and fraction of local Type BC particles and the abnormal decrease of its concentration and fraction during external aerosol transport. This explanation is adopted by previous research (Wen et al., 2021), where they also find the decrease of highly-oxygenated HULIS concentration in the urban area during the dust period."*

9. Page 15, line 405: Please, change 'ground-based-lidar' by 'ground-based wind lidar'.

Thank you for your suggestion, the word here is changed to a more formal name 'ground-based coherent Doppler wind lidar'

**Changes: line 539-541**

*"The combination of UV-LIF based online measurement instruments and ground-based coherent Doppler wind lidar expands the aerosol monitoring parameters and proves to be a potential method for real-time monitoring of fluorescent biological aerosol transport events."*

10. Page 20, Figure 1 (f): Please, use the same vertical scale for PM2.5 and PM10.

Corrected as suggested.

**Changes: line 781-788 Figure 1**

"

[Figure]

*Fig. 1 Left panel: Time-height crosssection of (a) attenuated backscatter coefficient, (b) Doppler spectral width, (c) horizontal wind speed, (d) horizontal wind direction, and (e) vertical wind speed over Hefei observed by CDWL on 13 March 2020. Right panel: simultaneous ground observation results, including (f) surface hourly particulate matter concentration from local monitor network, (g) number concentration and (h) size distribution of surface total aerosol particles from WIBS, (i) local temperature and humidity, and (j) local visibility and rain rate. The wind direction near the surface changes at about 03:30 and is marked with a triangle symbol.*

*,,*

11. Page 21, Figure 2 (a)-(j)-right panels: As far as possible, use the same vertical scale for Mean D and Mean AF.

Corrected as suggested.

**Changes: line 789-794 Figure 2**

"

[Figure]

Local Time (13 Mar 2020)

*Fig. 2 Left panel: number concentrations (solid red line), number fractions to total particles (blue chain line), and number fractions to fluorescent particles (black solid line with point marker) of the investigated particle types. Right panel: Count mean particle diameter (×5, solid red line) and count mean asphericity factor (blue chain line) of investigated particles measured by WIBS on 13 March 2020. The direction of the wind near the surface changes at about 03:30 and is marked with a triangle symbol.*

*,,*

12. Page 23, Figure 4 (f): Please, use the same vertical scale for PM2.5 and PM10.

Corrected as suggested.

**Changes: line 801-808 Figure 4**

*"*

[Figure]

*Fig. 4 Left panel: Time-height crosssection of (a) attenuated backscatter coefficient, (b) Doppler spectral width, (c) horizontal wind speed, (d) horizontal wind direction, and (e) vertical wind speed*

*over Hefei observed by CDWL between 16–17 March 2020. Right panel: simultaneous ground observation results, including (f) surface hourly particulate matter concentration from local monitor network, (g) number concentration and (h) size distribution of surface total aerosol particles from WIBS, (i) local temperature and humidity, and (j) local visibility and rain rate. The wind direction near the surface changes at about 01:30 on 17 March and is marked with a triangle symbol.*

"

13. Page 24, Figure 5 (a)-(j)-right panels: As far as possible, use the same vertical scale for Mean D and Mean AF.

Corrected as suggested.

Changes: line 809-814 Figure 5

"

[Figure]

Local Time (16 – 17 Mar 2020)

*Fig. 5 Left panel: number concentrations (solid red line), number fractions to total particles (blue chain line), and number fractions to fluorescent particles (solid black line with point marker) of investigated particle types. Right panel: Count mean particle diameter (×5, solid red line) and count mean asphericity factor (blue chain line) for investigated particle types measured by WIBS on 16–17 March 2020. The direction of the wind near the surface changes at about 01:30 on 17 March and is marked with a triangle symbol.*

*,,*

14. Page 26, Figure 7 (f): Please, use the same vertical scale for PM2.5 (x 10-1) and PM10.

Corrected as suggested.

**Changes: line 821-828, Figure 7**

*"*

[Figure]

*Fig. 7 Left panel: Time-height crosssection of (a) attenuated backscatter coefficient, (b) Doppler spectral width, (c) horizontal wind speed, (d) horizontal wind direction, and (e) vertical wind speed*

*over Hefei observed by CDWL on 19 March 2020. Right panel: simultaneous ground observation results, including (f) surface hourly particulate matter concentration from local monitor network ( $PM_{10} \times 10^{-1}$ for readability), (g) number concentration and (h) size distribution of surface total aerosol particles from WIBS, (i) local temperature and humidity, and (j) local visibility and rain rate. The wind direction near the surface changes at about 05:00 on 19 March and is marked with a triangle symbol.*

,,

15. Page 27, Figure 8 (a)-(j)-right panels: As far as possible, use the same vertical scale for Mean D and Mean AF.

Corrected as suggested.

**Changes: line 829-834, figure 8**

"

[Figure]

*Fig. 8 Left panel: number concentrations (solid red line), number fractions to total particles (blue chain line), and number fractions to fluorescent particles (solid black line with point marker) of investigated particle types. Right panel: Count mean particle diameter (×5, solid red line) and count mean asphericity factor (blue chain line) of investigated particle types measured by WIBS on 19 March 2020. The wind direction near the surface changes at about 05:00 on 19 March and is marked with a triangle symbol.*

*,,*

16. In Figures 3, 6 and 9, add more information (web, …) for Natural Earth, where the map data are from.

Additional information of NaturalEarth (naturalearthdata.com) is added in text and capitations of figures.

**Changes: line 229-230**

[revised manuscript text omitted]